# Do Vision-Language Models Really Understand Visual Language?

## Abstract

Visual language is a system of communication that conveys information through symbols, shapes, and spatial arrangements. Diagrams are a typical example of a visual language depicting complex concepts and their relationships in the form of an image. The symbolic nature of diagrams presents significant challenges for building models capable of understanding them. Yet, recent studies seem to suggest that Large Vision-Language Models (LVLMs) can even tackle complex reasoning tasks involving diagrams. In this paper, we investigate this phenomenon by developing a comprehensive test suite to evaluate the diagram comprehension capability of LVLMs. Our test suite uses a variety of questions focused on concept entities and their relationships over a set of synthetic as well as real diagrams across several domains to evaluate the recognition and reasoning abilities of models. Our evaluation of three LVLMs (GPT-4V, GPT-4o, and Gemini) shows that while these models can accurately identify and reason about entities, their ability to understand relationships is notably limited. Further testing reveals that the decent performance on diagram understanding largely stems from leveraging their background knowledge as shortcuts to identify and reason about the relational information. Thus, we conclude that LVLMs have a limited capability for genuine diagram understanding, and their impressive performance in diagram reasoning is an illusion emanating from other confounding factors, such as the background knowledge in the models.

## 1 Introduction

Symbolic signals such as language serve as powerful tools in communication by abstracting and interpreting information. Visual language is a form of communication that uses symbols, shapes, and spatial arrangements to convey complex ideas (Greenspan & Shanker, 2009; Li, 2023). Diagrams, which encapsulate symbolic information in the visual stream, are a prime example of visual language (Zdebik, 2012; Anderson et al., 2011) that are extensively used in practice across various domains, e.g., mathematics (Seo et al., 2015), science (Lu et al., 2022), education (Kembhavi et al., 2016; 2017), and illustrations (Hiippala & Orekhova, 2018; Lu et al., 2021). Developing models capable of understanding symbolic information, e.g. in diagrams, is a critical milestone in advancing machine intelligence (Bauer & Johnson-Laird, 1993; de Rijke, 1999; Cromley et al., 2010). Even though recent Large Vision-Language Models (LVLMs, OpenAI, 2023; Anil et al., 2023) have demonstrated some success on diagram-based visual reasoning tasks (Lu et al., 2023; Zhang et al., 2024; Chen et al., 2024), it remains unclear whether the performance on these tasks truly reflects the models' ability to comprehensively understand the symbolic information in diagrams.

For this purpose, we design a comprehensive test suite that investigates the ability of LVLMs to understand diagrams. As defined by Foucault (1977) and Deleuze (1986), diagrams are abstract tools that organize visual entities using relational information. Drawing inspiration from this, our test suite focuses on evaluating diagram understanding by assessing how well models can understand entities and relations in typical diagrams (§ 2.1). We evaluate diagram understanding by defining two types of tasks pertaining to fast recognition of entities and relations and slow multi-step reasoning (Kahneman, 2011) over the relations (§ 2.2). While we cannot cover every diagram type for practical reasons, we still cover diagrams across six domains. To ensure that our evaluation is both controlled and generalizable, our test suite includes both clean synthetic diagrams and $1,001$ annotated real diagrams carefully selected from existing datasets Krishnamurthy et al. (2016); Kembhavi

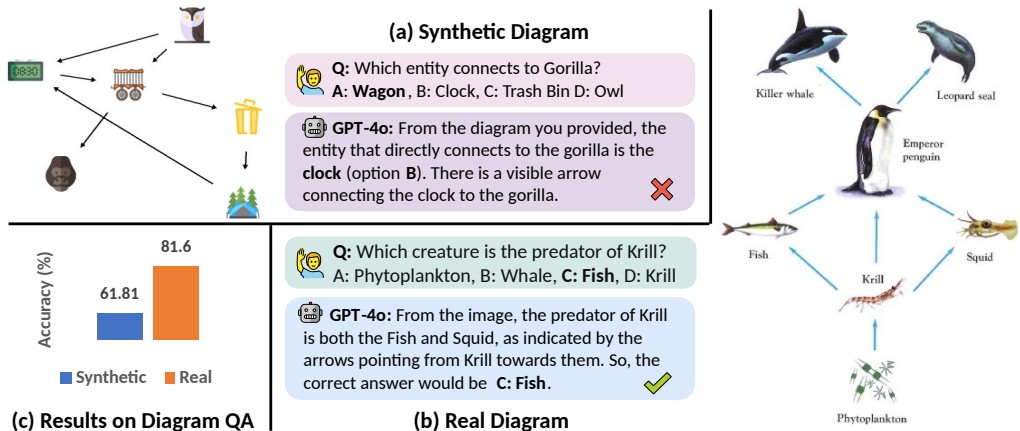

Figure 1: The responses of GPT-4o to two diagram-related questions reveal a notable pattern. The model struggles to correctly answer the relation question in the simple synthetic diagram, yet it successfully understands the relationship in a complex real diagram. We demonstrate that this pattern occurs consistently (Tabs. 4 and 5).

et al. (2016) (§ 2.3). Using our test suite, we conduct a detailed analysis of the model's strengths and weaknesses in understanding diagrams and explore the following questions:

**Q1: Can LVLMs understand diagrams?** To assess the basic capabilities of LVLMs, we first test them using clean synthetic diagrams, followed by evaluations in real-world scenarios for comparison. Our findings from these evaluations lead to three key observations:

- *LVLMs can identify entities and reason about them.* By generating synthetic diagrams to evaluate models from multiple perspectives, we observe that models consistently perform well on entity-related questions. They can accurately identify and reason about entities in synthetic diagrams, regardless of whether the entity is represented textually or visually (§ 3.1).

- *LVLMs struggle with identifying and reasoning about relations (Fig. 1a).* In synthetic scenarios, the models exhibit significant difficulty in identifying relationships between depicted concepts and in performing reasoning tasks based on those relationships (§ 3.2). This challenge persists across various diagram settings and prompting templates (Apps. E.2.2 and E.2.3).

- *For real diagrams, LVLMs still understand entities and cannot reason about relations. But they can identify relations (Fig. 1b).* We annotate real diagrams with questions from multiple aspects and evaluate models on them. Unexpectedly, we find that the models perform significantly better on relation recognition questions in real diagrams compared to those in synthetic diagrams (§ 3.3).

**Q2: If LVLMs cannot identify relations in simple synthetic diagrams, how do they manage to answer complex questions in practice? (Fig. 1c)** One potential hypothesis is that the models leverage their background knowledge as a shortcut to answer these questions. To test it, we explore the impact of knowledge on question-answering (QA) and draw the following observations:

- *Knowledge enhances model performance in relation recognition.* We construct synthetic diagrams that incorporate semantic knowledge and observe a notable improvement in relation recognition questions, suggesting that models perform better on knowledge-grounded diagrams (§ 4.1). Additionally, in real diagrams, we categorize questions based on whether they require background knowledge (e.g., commonsense) or can be answered independently of it. The results show that LVLMs excel at answering questions that draw upon background knowledge (§ 4.2).

- *LVLMs only answer relation questions correctly for simple real diagrams.* We classify diagrams by complexity that is based on the number of entities, and analyze QA performance for simple and complex diagrams. Results reveal that while accuracy on entity questions remains consistent, that on relation questions drops significantly with the increase of complexity. This indicates that the models' seemingly good performance on relation questions is primarily driven by handling simpler diagrams, rather than by genuine relation understanding (§ 5.1).

- *LVLMs rely on learned knowledge to hallucinate relations.* In the case study, we demonstrate that even when no relations are provided, LVLMs infer relations based on their learned knowl-

edge. Furthermore, if provided relations contradict the models' learned knowledge, they tend to disregard them and instead rely on their background knowledge to answer the questions (§ 5.2).

**Findings.**   We summarize our research findings as follows: While LVLMs can recognize and reason about entities, they struggle with relations. The models do not engage in genuine diagram parsing or reasoning; rather, their seemingly strong performance on various diagram benchmarks is an illusion created by their reliance on ***knowledge shortcuts***. Specifically, ***these models identify the entities depicted in the diagrams and simply retrieve relevant pre-learned knowledge***.

## 2   TEST SUITE DESIGN

In this section, we provide a definition of a diagram and provide a desiderata for our evaluation suite.

### 2.1   DIAGRAMS AS GRAPHS

Diagrams work as an abstract tool to describe concepts and relationships (Foucault, 1977; Deleuze, 1986). In practice, we choose this representation as it is quite flexible and a broad set of diagrams can be represented in a format as shown in Fig. 1. For example, logical diagrams such as water cycles illustrate the process transitions (relations) among cycle stages (entities). Schematic diagrams such as circuits demonstrate the connections (relations) among electronic components (entities). Previous work (Song et al., 1995; Hiippala & Orekhova, 2018) also chose to annotate and model diagrams as concepts and their connections.[1]

Following this representation, we define a diagram as a graph $G = \{\mathcal{V}, \mathcal{E}\}$. Here, $\mathcal{V}$ is the set of entities (e.g., "Squid" in the example diagram in Fig. 1b). Each entity $V \in \mathcal{V}$ could be represented in multiple ways, e.g., text and visuals in the diagram. Each relation $E = (V, V') \in \mathcal{E}$ connects two entities. Relations are either explicitly represented by arrows or via implicit relationships (e.g., relative positioning of entities).

### 2.2   HOW DO WE EVALUATE THE MODEL'S DIAGRAM UNDERSTANDING ABILITY?

In designing our test suite, we draw inspiration from  (Kahneman, 2011) who argues that the thinking process can be naturally divided into two modes: System 1, which handles automatic, quick, and intuitive thinking (e.g., pattern recognition and everyday decisions), and System 2, which is responsible for deliberate, slow, and logical thinking (e.g., logical reasoning and critical analysis). We evaluate both the recognition and reasoning abilities of models via question-answering (QA). We carefully design a set of questions posed in a multiple-choice QA format with each question having one correct answer and three incorrect options and simply use the model's accuracy in answering the questions as an evaluation of the model's ability on that skill. Overall, we denote the set of questions as $Q$. We denote the set of questions related to understanding entities as $Q(V)$, while those related to relations as $Q(E)$. Additionally, we use subscripts to distinguish between different types of questions: questions on synthetic diagrams are denoted as $Q_S$, whereas questions for real diagrams are denoted as $Q_R$. Our test suite questions are categorized as follows:

**Recognition vs. Reasoning Questions.**   To measure the two key abilities of models in recognition of entities and relations vs. reasoning, we design two types of questions: Name Recognition (NR) and Number Counting (NC). Specifically, the NR questions measure the recognition ability of models by verifying the existence of specific entities or relations. In contrast, NC questions measure reasoning ability by asking for the number of certain types of entities or relations. We formally denote these question sets as $Q(\cdot|\text{NR})$ and $Q(\cdot|\text{NC})$, respectively.

**Knowledge-Required vs. Knowledge-Free.**   Next, we test if LVLMs use any knowledge shortcuts (Ye & Kovashka, 2021; Tang et al., 2023) to answer our questions without true diagram understanding. Diagrams often encode some background real-world knowledge, and the models may use their background knowledge as a shortcut to answer the questions. To further tease out the models'

---

[1]Given the vast number of diagram types, we would leave certain complex cases that are challenging to be represented in this way for future work.

true understanding of diagrams, we design questions that both do and do not require background knowledge, allowing us to test the models' capabilities in each scenario. If a question requires the model to use background knowledge (e.g., semantic or commonsense knowledge), we call it a knowledge-required (KR) question, which is denoted as $Q(\cdot|\text{KR})$. On the other hand, questions that do not rely on such external knowledge are termed knowledge-free (KF) questions, denoted as $Q(\cdot|\text{KF})$. This distinction helps us clearly separate the model's capacity for pure visual processing from its ability to incorporate and utilize prior knowledge when answering questions.

| Synthetic Diagram | Question | Template |
|---|---|---|
| Entity | $Q_S(V|\text{KF, NR})$ | Which one of the entities exists in the diagram? |
|  | $Q_S(V|\text{KF, NC})$ | How many text labels are there in the diagram? |
| Implicit Relation | $Q_S(E|\text{KF, NR})$ | Which one of the text labels is placed on the left of the entity cow? |
|  | $Q_S(E|\text{KF, NC})$ | How many text labels are placed on the left of the entity cow? |
| Explicit Relation | $Q_S(E|\text{KF, NR})$ | Which one of the pairs is connected in the diagram? |
|  | $Q_S(E|\text{KF, NC})$ | How many entities are connected to cow? |

| Real Diagram | Question | Example |
|---|---|---|
| Entity | $Q_R(V|\text{KF, NR})$ | How many entities are there in the diagram? |
|  | $Q_R(V|\text{KF, NC})$ | Which entity is in the diagram? |
|  | $Q_R(V|\text{KR, NR})$ | Which producer is in the diagram? |
|  | $Q_R(V|\text{KR, NC})$ | How many consumers are in the diagram? |
| Relation | $Q_R(E|\text{KF, NR})$ | Which entity is connected to Fish? |
|  | $Q_R(E|\text{KF, NC})$ | How many arrows are linked to Fish in the diagram? |
|  | $Q_R(E|\text{KR, NR})$ | Which is not the predator of Krill? |
|  | $Q_R(E|\text{KR, NC})$ | How many types of prey are consumed by Fish in the foodweb? |

Table 1: The template and example of question annotations. The underlined entity varies across diagrams. There are no KR questions for synthetic diagrams since they do not have background knowledge. Questions for real diagrams correspond to Fig. 1b. In terms of relations, "Explicit Relation" refers to relations that are clearly depicted through arrows or segments, while "Implicit Relation" refers to those that are conveyed indirectly, such as through relative position relationships.

Templates and examples of each question type are presented in Tab. 1. Each question type targets a specific diagram component or model ability within the evaluation. For real diagrams, the question templates are tailored to the specific context of each domain. Details on the question design in each domain are in App. D.1. From these questions, we can derive the following intuition:

**Intuition 1.** *KR questions in real diagrams are generally more challenging than KF questions in synthetic diagrams. The reasons are: 1) Real diagrams are inherently more complex, containing a wider range of information compared to synthetic diagrams; 2) Beyond assessing basic abilities, answering KR questions also requires the integration of additional background knowledge.*

## 2.3 WHICH DIAGRAMS DO WE CONSIDER?

Diagrams are extensively utilized in various domains, appearing in different forms and encompassing a wide range of information types. While evaluating models in a synthetic setting helps to reduce the impact of confounding factors, the resulting conclusions may not fully extend to real-world cases. Conversely, evaluating models on real diagrams allows for broader coverage of diagram types, though it may introduce biases due to the additional information or knowledge embedded in these diagrams. To address this challenge, our test suite incorporates both synthetic and real diagrams, providing a balanced and comprehensive evaluation.

**Synthetic diagram set.** We generate synthetic diagrams in two steps. First, we randomly create between 2 to 9 entities represented by images or text. Then, we randomly establish between 1 to maximum relations among them using directed arrows.[2] To ensure clarity in the synthetic diagrams, we carefully avoid situations when arrows cross over certain entities. In practice, we construct our

---

[2]The probability of random strategy distributes uniformly across entity number and relation number.

entity set $\mathcal{V}$ by sampling from a pre-defined set containing $377$ distinct entities provided by Lu et al. (2021). By default, we generate $1,000$ diagrams for each experiment.

| Domains | Ecology | Biology | Physics | Astronomy | Chemistry | Geology |
|---|---|---|---|---|---|---|
| Num. | 462 | 205 | 77 | 101 | 54 | 102 |
| Entity Rep. | Text & Visual | Text & Visual | Visual | Text & Visual | Text & Visual | Text & Visual |
| Relation Rep. | Explicit | Explicit | Explicit | Implicit | Explicit | Implicit |
| Topics | Food Chain Food Web | Life Cycle | Circuit | Solar System Satellite System | Water Cycle Carbon Cycle | Planet Structure Star Structure |

Table 2: Details of the real diagram set. "Entity Rep." and "Relation Rep." refer to the way that entities and relations are represented. "Topics" introduces the typical types of diagrams in that domain.

**Real diagram set.** We carefully filtered and curated a selection of $1,001$ real-world diagrams from Krishnamurthy et al. (2016); Kembhavi et al. (2016) to include in our test suite. These diagrams span a diverse range of domains, including ecology, biology, physics, astronomy, chemistry, and geology. This selection ensures that our test suite covers a broad spectrum of scientific disciplines, providing a comprehensive evaluation of the models' capabilities. During the filtering process, the diagrams were first categorized by domain. Subsequently, low-quality diagrams, along with those considered too simplistic or excessively complex, were removed to ensure reliable annotations. Detailed statistical information about these diagrams can be found in Tab. 2. Example questions are given in Tab. 1.

## 3 DO LVLMS UNDERSTAND DIAGRAMS?

In this section, we investigate whether LVLMs can understand entities (§ 3.1) and relations (§ 3.2) in synthetic diagrams. Additionally, we present the evaluation results on real diagrams (§ 3.3).

### 3.1 CAN LVLMS UNDERSTAND ENTITIES?

We begin by evaluating whether LVLMs can identify and reason about entities represented by text boxes (i.e., text entities) or visual icons (i.e., visual entities). Additionally, we examine the models' ability to correctly identify the spatial information (i.e., locations) of these entities in App. E.2.1.

**Preparation.** We evaluate three LVLMs: GPT-4Vision (i.e., GPT-4V, OpenAI, 2023)), GPT-4o (OpenAI, 2024), and Gemini 1.5 Pro (i.e., Gemini, Anil et al., 2023)), where the evaluation takes place from June to September 2024. Details about model configurations are in App. E.1. Prompting templates and demonstration examples for various models are given in Figs. 11 to 14 in App. F.1.1.

| Accuracy (%) | Text Entity | | Visual Entity | |
|---|---|---|---|---|
| | $Q_S(V|\text{KF}, \textbf{NR})$ | $Q_S(V|\text{KF}, \textbf{NC})$ | $Q_S(V|\text{KF}, \textbf{NR})$ | $Q_S(V|\text{KF}, \textbf{NC})$ |
| GPT-4V (ZS / CoT) | 97.41 / 97.81 | 50.60 / 99.60 | 83.40 / 85.74 | 32.38 / 93.65 |
| GPT-4o (ZS / CoT) | 91.63 / 99.20 | 64.06 / 100.0 | 87.50 / 92.58 | 46.72 / 94.88 |
| Gemini (ZS / CoT) | 86.85 / 88.05 | 71.89 / 95.78 | 90.23 / 87.70 | 67.21 / 86.48 |
| **Average (ZS / CoT)** | 91.97 / 95.02 | 62.18 / 98.46 | 87.04 / 88.67 | 48.77 / 91.67 |

Table 3: Performance of LVLMs on QA in terms of entities in text boxes or visual icons. LVLMs can always identify entities correctly, and can also reason about them effectively with CoT prompting.

**Results.** We evaluate LVLMs under two settings: zero-shot prompting (ZS) and the Chain-of-Thought prompting (CoT, Wei et al., 2022) as in Tab. 3. The results demonstrate that all LVLMs can easily recognize entities in both text boxes ($> 95\%$ accuracy) and visual icons ($> 88\%$ accuracy). The accuracies on NR questions, which assess entity recognition, remain consistently high. For the NC questions, which evaluate reasoning ability, we find that LVLMs can answer almost perfectly with CoT prompting, achieving $98.46\%$ accuracy for text entities and $91.67\%$ for visual entities. Our findings on entity recognition and reasoning can be summarized as follows:

**Observation 1** (Ability to understand entity). *LVLMs can nearly perfectly identify entities in diagrams and demonstrate strong reasoning abilities regarding these entities.*

This ability is fundamental to various vision tasks. Our observation aligns with existing research, confirming that models possess the basic capability to perform simple object detection and count objects to some extent. Next, we turn our focus to complex relations to determine whether LVLMs can comprehend the intricate interactions between entities.

## 3.2 CAN LVLMs UNDERSTAND RELATIONS?

We categorize relations into two types for our research: implicit relations (e.g., relative positions of entities) and explicit relations (e.g., arrows or segments).

**Preparation.** We generate synthetic diagrams following previous settings (§ 3.1). To reduce errors contributed by entity understanding, here we represent entities by text, which yields the best performance on corresponding NR and NC questions (Tab. 3). Example questions are in Tab. 1. The prompting templates and demonstration examples are in Figs. 17 to 20 in App. F.1.2.

| Accuracy (%) | Implicit Relation | | Explicit Relation | |
| --- | --- | --- | --- | --- |
| | $Q_S(E|\text{KF}, \text{NR})$ | $Q_S(E|\text{KF}, \text{NC})$ | $Q_S(E|\text{KF}, \text{NR})$ | $Q_S(E|\text{KF}, \text{NC})$ |
| GPT-4V (ZS / CoT) | 75.89 / 72.33 | 30.36 / 34.41 | 57.59 / 61.60 | 49.62 / 59.51 |
| GPT-4o (ZS / CoT) | 72.53 / 77.27 | 37.04 / 55.26 | 61.81 / 76.58 | 57.60 / 70.15 |
| Gemini (ZS / CoT) | 58.50 / 60.87 | 30.36 / 31.78 | 60.97 / 68.52 | 69.58 / 70.15 |
| **Average (ZS / CoT)** | 68.97 / 70.16 | 32.59 / 40.49 | 60.13 / 68.00 | 58.94 / 66.60 |

Table 4: Performance of LVLMs on QA for relations. LVLMs struggle to identify both implicit and explicit relations and are unable to reason about them effectively.

**Results.** Tab. 4 presents the accuracies for relation questions. The results indicate that all models generally struggle with relation recognition (NR questions), even when using CoT prompting, which leads to an average accuracy of around 70%. Similarly, the models also have difficulty reasoning about relations (NC questions), with the average accuracy ranging from 40% to 66%. Notably, GPT-4V's performance on counting implicit spatial relations (i.e., relative positions) is nearly equivalent to random guessing (34.41%), and it shows significant difficulty in recognizing or reasoning about explicit relations, even under the CoT prompting setting.

**Consistency Verification.** Before proceeding, we validate our findings on relations to ensure their reliability. We adjust the diagram generation settings (e.g., arrow features) and observe that the results remained consistent (App. E.2.2). Beyond diagram variations, we also test the robustness of our results by examining the consistency of LVLMs' performance with different prompting templates, specifically under the in-context learning (ICL) setting. The findings indicate that ICL does not improve the models' ability to identify or reason about relations (App. E.2.3). These results further confirm the reliability of our conclusions. Thus, we can summarize our observation as follows:

**Observation 2** (Ability to understand relations). *LVLMs can barely identify both implicit and explicit relations, and they are unable to reason about them, even with CoT prompting.*

This observation contradicts the remarkable success that LVLMs have demonstrated in understanding complex diagrams. To further investigate their failures, we evaluate them on real diagrams to determine whether they can effectively comprehend more complex, real-world scenarios.

## 3.3 DO LVLMs UNDERSTAND REAL DIAGRAMS?

Synthetic diagrams are used to evaluate the basic abilities of models. Next, we move to real diagrams to double-check how these well-trained models perform in practical diagram understanding.

**Preparation.** The evaluation follows the similar settings of that on synthetic diagrams. We first provide the performance on KR questions for both entities and relations in real diagrams. Example questions are given above (Tab. 1), and the question design for diagrams in each domain is in App. D.1. Prompt templates and examples can be found in Figs. 27 to 30 in App. F.2.

| Accuracy (%) | Entity | | Relation | |
|---|---|---|---|---|
| | $Q_R(V\|\text{KR}, \text{NR})$ | $Q_R(V\|\text{KR}, \text{NC})$ | $Q_R(E\|\text{KR}, \text{NR})$ | $Q_R(E\|\text{KR}, \text{NC})$ |
| GPT-4V (ZS / CoT) | 84.30 / 88.89 | 49.41 / 78.77 | 77.48 / 78.68 | 42.10 / 59.89 |
| GPT-4o (ZS / CoT) | 87.90 / 93.10 | 61.40 / 82.29 | 81.60 / 84.10 | 50.40 / 72.89 |
| Gemini (ZS / CoT) | 87.29 / 84.99 | 53.49 / 68.35 | 80.90 / 80.50 | 51.39 / 57.69 |
| **Average (ZS / CoT)** | 86.49 / 88.99 | 54.77 / 76.47 | 79.99 / 81.09 | 47.96 / 63.49 |

Table 5: Performance of LVLMs on KR questions for real diagrams. Results indicate that models continue to recognize and reason about entities effectively, and they struggle with reasoning about relations. However, surprisingly, they are able to recognize relations in real diagrams.

**Results.** Tab. 5 presents the performance of LVLMs on real diagrams. We observe that models continue to perform well in recognizing and reasoning about entities, with GPT-4o achieving 93.10% accuracy in recognition and 82.29% accuracy in reasoning. Besides, models still struggle to reason about relations in real diagrams, similar to their performance on synthetic diagrams. Notably, though, we find that models can recognize relations quite well in real diagrams, with an average accuracy of 81.09%, compared to only 68.00% on synthetic diagrams (Tab. 4). From these results, we can obtain the following observation:

**Observation 3** (Performance on real diagrams). *LVLMs struggle to recognize relations in simple synthetic diagrams, yet they can effectively recognize relations in complex real diagrams.*

We substantiate that LVLMs cannot understand relations in synthetic diagrams, yet this finding reveals a contradiction. While the models do not inherently possess the ability to recognize relations, they are able to do so in real diagrams. This outcome contradicts our initial intuition (**Intuition** 1). Therefore, we further investigate these counterintuitive findings by examining the key difference between synthetic and real diagrams: the role of knowledge. While synthetic diagrams contain entities and relations that are random in their construction, real diagrams often portray entities and relations that agree with commonsense knowledge about the underlying concepts —such as the stages of the water cycle or the predator-prey relations in a food chain.

## 4    KNOWLEDGE AS SHORTCUTS: QUANTITATIVE ANALYSIS

Given that rich knowledge has been encoded into LVLMs during various training stages, a possible *hypothesis* emerges: **these models may not truly understand diagrams but instead rely on their ingrained knowledge as shortcuts to provide answers**. In this section, we explore the impact of knowledge on the models' ability to answer questions.

### 4.1    KNOWLEDGE GROUNDING OF DIAGRAMS IMPROVES RELATION RECOGNITION

To determine the effect of knowledge, we first compare the models' QA performance on diagrams with and without embedded knowledge. Our focus is on the explicit relations in synthetic diagrams. We construct relations that incorporate semantic knowledge to simulate practical conditions where models might use this knowledge as shortcuts. If our hypothesis is correct, we would expect to see an increase in accuracy on QA tasks for these specially constructed diagrams.

**Preparation.** To construct diagrams grounded with semantic knowledge, we generate relations with real meaning behind them. Specifically, for each entity, we get its Word2Vec embedding (Mikolov et al., 2013) based on the text attribute, and use cosine similarity implemented by spaCy (Honnibal et al., 2020). If the similarity between entity text is larger than 0.5, we regard that there exists a relation. We construct a *semantic graph* on all the entities from Lu et al. (2021) with 377 entities and 1901 relations. We randomly generate diagrams following the settings mentioned

in § 2.3, but we constrain the generated diagram as the subgraph in the constructed *semantic graph*.[3] See Figs. 25 and 26 in App. F.1.5 for the prompting templates and demonstration examples.

| Accuracy (%) | $Q_S(E|$KF, NR$)$ | | $Q_S(E|$KF, NC$)$ | |
| Diagram Type | Vanilla | Knowledge-Grounded$_{\Delta\uparrow\downarrow}$ | Vanilla | Knowledge-Grounded$_{\Delta\uparrow\downarrow}$ |
|---|---|---|---|---|
| GPT-4V (ZS / CoT) | 57.59 / 61.60 | $74.17_{16.58\uparrow}$ / $74.39_{12.79\uparrow}$ | 49.62 / 59.51 | $55.79_{6.17\uparrow}$ / $57.87_{1.64\downarrow}$ |
| GPT-4o (ZS / CoT) | 61.81 / 76.58 | $77.92_{16.11\uparrow}$ / $82.78_{6.20\uparrow}$ | 57.60 / 70.15 | $60.65_{3.05\uparrow}$ / $72.92_{2.77\uparrow}$ |
| Gemini (ZS / CoT) | 60.97 / 68.52 | $72.19_{11.22\uparrow}$ / $72.19_{6.37\uparrow}$ | 69.58 / 70.15 | $68.06_{1.52\downarrow}$ / $64.81_{5.34\downarrow}$ |
| **Average (ZS / CoT)** | 60.13 / 68.00 | $75.57_{15.44\uparrow}$ / $76.45_{8.45\uparrow}$ | 58.94 / 66.60 | $61.50_{2.56\uparrow}$ / $65.20_{1.40\downarrow}$ |

Table 6: Performance on synthetic diagrams without and with semantic knowledge (denoted as "Vanilla" and "Knowledge-Grounded"). The accuracy change from Vanilla to Knowledge-Grounded reveals that models better identify relations in knowledge-grounded diagrams while still struggling to reason about them effectively.

**Results.** Tab. 6 presents the evaluation accuracies on diagrams without and with semantic knowledge. For comparison, we include the performance changes relative to the original results shown in Tab. 4. Overall, models exhibit improved relation recognition in diagrams containing semantic knowledge. The average accuracy improvement for NR questions is $15.44\%$ with zero-shot (ZS) prompting and $8.45\%$ with CoT prompting. However, for NC questions, the performance remains largely unchanged. The findings are consistent with our hypothesis: models utilize knowledge in diagrams as relation recognition shortcuts. With these results, we have below observation:

**Observation 4** (Knowledge in diagrams helps). *LVLMs are more effective at recognizing relations in diagrams that relate to some background knowledge, as they can use it as a shortcut.*

This observation indicates that even when questions do not explicitly require background knowledge to answer them, the presence of knowledge in the diagram can still enhance the performance of relation recognition. Next, we investigate whether questions that require models to actively use knowledge could further improve their performance.

## 4.2 RELATION RECOGNITION QUESTIONS REQUIRING KNOWLEDGE SHOW IMPROVEMENTS

We then evaluate the models using questions that do not require background knowledge (KF questions) and compare their performance with questions that do require such knowledge (KR questions).

| Accuracy (%) | $Q_R(V|$KF, NR$)$ | $Q_R(V|$KR, NR$)_{\Delta\uparrow\downarrow}$ | $Q_R(V|$KF, NC$)$ | $Q_R(V|$KR, NC$)_{\Delta\uparrow\downarrow}$ |
|---|---|---|---|---|
| GPT-4V (ZS / CoT) | 91.90 / 91.50 | $84.30_{7.60\downarrow}$ / $88.89_{2.61\downarrow}$ | 38.59 / 75.24 | $49.41_{10.81\uparrow}$ / $78.77_{3.53\uparrow}$ |
| GPT-4o (ZS / CoT) | 88.30 / 93.60 | $87.90_{0.40\downarrow}$ / $93.10_{0.51\downarrow}$ | 45.89 / 79.05 | $61.40_{15.51\uparrow}$ / $82.29_{3.24\uparrow}$ |
| Gemini (ZS / CoT) | 88.10 / 82.00 | $87.29_{0.81\downarrow}$ / $84.99_{2.99\uparrow}$ | 58.38 / 75.46 | $53.49_{4.90\downarrow}$ / $68.35_{7.12\downarrow}$ |
| **Average (ZS / CoT)** | 89.43 / 89.03 | $86.49_{2.94\downarrow}$ / $88.99_{0.04\downarrow}$ | 47.62 / 76.58 | $54.77_{7.14\uparrow}$ / $76.47_{0.11\downarrow}$ |

Table 7: The results (e.g., accuracy change from answering KF questions to KR questions) indicate that models perform similarly on entity questions, regardless of whether or not they require background knowledge.

**Evaluation on entity questions** We follow the same settings as in previous experiments. Tab. 7 illustrates the role of knowledge in questions related to entities. The results show that the gap between KF and KR questions for entities in real diagrams is negligible. Specifically, the average accuracy decrease in recognition is only $0.04\%$ under the CoT setting, while the decrease in reasoning is similarly minimal at $0.11\%$.

**Evaluation on relation questions.** Similarly, Tab. 8 illustrates the impact of knowledge on questions related to relations. The results show that when questions require knowledge, models are better at recognizing relations. However, their reasoning ability remains largely unchanged. Specifically, the average accuracy gap in recognition is $11.13\%$ under the CoT setting, while the gap in reasoning is only $2.4\%$. Thus, we have the observation below:

---

[3]Under such strict constraints, we can only generate 885 (instead of $1,000$) distinct diagrams for evaluation.

| Accuracy (%) | $Q_R(E\|\textbf{KF}, \text{NR})$ | $Q_R(E\|\textbf{KR}, \text{NR})_{\Delta\uparrow\downarrow}$ | $Q_R(E\|\textbf{KF}, \text{NC})$ | $Q_R(E\|\textbf{KR}, \text{NC})_{\Delta\uparrow\downarrow}$ |
|---|---|---|---|---|
| GPT-4V (ZS / CoT) | 66.99 / 69.69 | $77.48_{10.49\uparrow}$ / $78.68_{8.99\uparrow}$ | 38.30 / 57.09 | $42.10_{3.81\uparrow}$ / $59.89_{2.80\uparrow}$ |
| GPT-4o (ZS / CoT) | 69.20 / 73.90 | $81.60_{12.39\uparrow}$ / $84.10_{10.20\uparrow}$ | 49.97 / 65.00 | $50.40_{0.43\uparrow}$ / $72.89_{7.89\uparrow}$ |
| Gemini (ZS / CoT) | 68.00 / 66.30 | $80.90_{12.90\uparrow}$ / $80.50_{14.20\uparrow}$ | 53.39 / 61.18 | $51.39_{2.00\downarrow}$ / $57.69_{3.49\downarrow}$ |
| **Average (ZS / CoT)** | 68.07 / 69.96 | $79.99_{11.93\uparrow}$ / $81.09_{11.13\uparrow}$ | 47.22 / 61.09 | $47.96_{0.75\uparrow}$ / $63.49_{2.40\uparrow}$ |

Table 8: Model performance for questions on relations in real diagrams. The accuracy gap (from answering KF questions to KR questions) suggests that models perform better on NR questions when these questions require knowledge to answer. However, there is no significant improvement in accuracy when it comes to NC questions.

**Observation 5** (Knowledge shortcuts help). *LVLMs are better at recognizing relations in real diagrams when the question requires them to use background knowledge. However, whether or not a question requires knowledge does not significantly impact the models' performance in entity recognition, entity reasoning, or reasoning about relations.*

These two observations (**Observations** 4 and 5) lead to conclusions that are entirely contrary to our initial intuition (**Intuition** 1). In this section, we use quantitative analysis to support our hypothesis that knowledge acts as a shortcut for recognizing relations. In the next section, we provide qualitative analysis to further substantiate the validity of this hypothesis.

## 5 KNOWLEDGE SHORTCUTS: QUALITATIVE ANALYSIS

We provide further evidence to confirm that LVLMs can only recognize and reason about entities but not relations in real diagrams (§ 5.1). The ability to recognize relations in real diagrams appears to be an illusion driven by knowledge shortcuts rather than genuine understanding (§ 5.2).

### 5.1 LVLMs CANNOT RECOGNIZE AND REASON ABOUT RELATIONS IN REAL DIAGRAMS

**Preparation.** We use the number of entities in a diagram as an indicator of its complexity, with the answers to $Q_R(V|\text{KF}, \text{NC})$ providing the entity count, as introduced in Tab. 1. We then divide all real diagrams into five bins based on their entity count, ensuring that each bin contains more than 100 diagrams (detailed statistics are provided in Fig. 9). For each subset of diagrams, we report the average accuracy for all questions under the CoT setting.

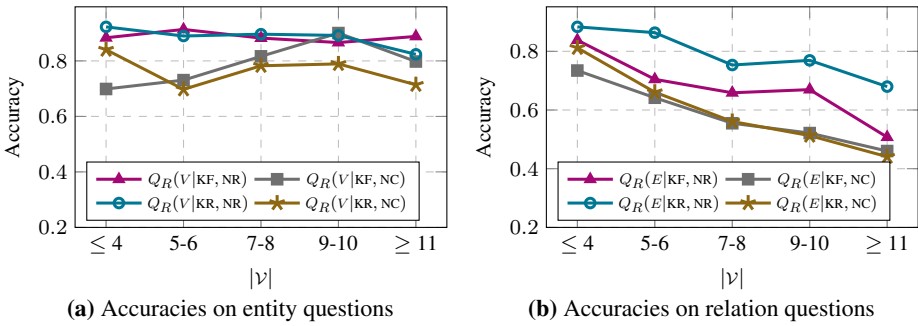

**(a)** Accuracies on entity questions    **(b)** Accuracies on relation questions

Figure 2: Performance of LVLMs (CoT) on answering questions for real diagrams with different complexities (i.e., the number of entities in the diagram, $|\mathcal{V}|$). Results show that models can always answer questions on entity well but cannot handle questions on relations if the diagram is complex.

**Results.** Fig. 2 present the average accuracy of three models on all questions (detailed results are in Fig. 10 in App. E.3). Overall, the performance on entity questions remains consistent across different levels of diagram complexity, while there is a noticeable decline in accuracy for relation questions as diagram complexity increases. These results further support that LVLMs can understand entities but struggle with understanding relations. Additionally, they suggest that the models' apparent success in recognizing and reasoning about relations in real diagrams is largely due to their performance on simpler diagrams, rather than a true comprehension of complex relational structures.

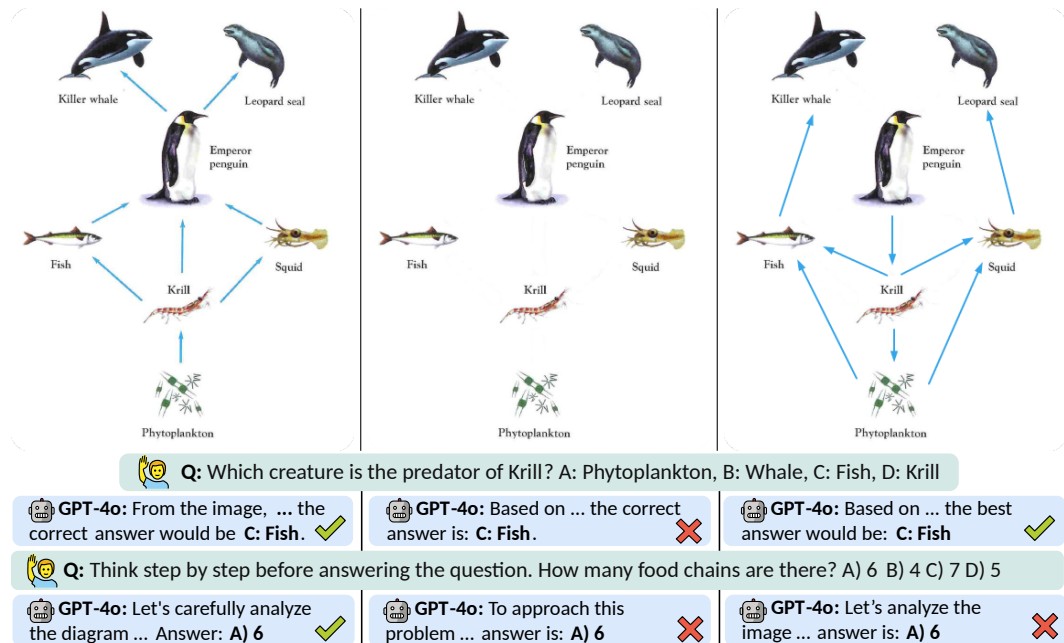

Figure 3: The model response on the example diagram and its variants. Results suggest that the model relies on background knowledge as a shortcut rather than accurately recognizing and reasoning about relations.

## 5.2 LVLMs Hallucinate Relations in Real Diagrams (Case Study)

Next, we provide an intuitive and vivid case study to demonstrate that LVLMs hallucinate when interpreting relations in diagrams. Using the example diagram from Fig. 1, we construct two special cases for comparison: in the first case, we remove all relations from the diagram, and in the second case, we replace the relations with random ones. The goal is to test the GPT-4o's response when there is either no relational information or when the relations presented conflict with background knowledge. For evaluation, we pose an annotated question ($Q_R(E|\text{KR}, \text{NR})$) and a complex reasoning question involving food chain counting and use the configuration in App. E.1.

In Fig. 3, when comparing the responses in the left subfigure (vanilla) with those in the middle subfigure (w/o relation), we observe that even in the absence of explicit relational information, the model still identifies the correct predator. Additionally, for the food chain counting question, the model continues to provide the original answers. This indicates that the model has pre-existing knowledge and it can use the knowledge as a shortcut for answering questions. Similarly, when comparing the responses in the left subfigure (vanilla) with those in the right subfigure (with random relations), we find that the model provides the same answers despite the introduction of random relations. The new correct predator could be "A)" or "C)", and the new correct food chain count is 4. This demonstrates that the model relies on learned knowledge rather than parsing the diagram itself. With this additional evidence, we can reasonably conclude the following finding:

**Finding:** Current LVLMs can recognize and reason about entities in diagrams but struggle with understanding relations. However, they manage to answer diagram-related complex questions by identifying entities and leveraging relevant learned knowledge as a shortcut.

## 6 Conclusion

We evaluate three LVLMs on diagram understanding using our test suite, including synthetic and real diagrams. Our findings reveal that while these models can perfectly recognize and reason about entities depicted in the diagrams, they struggle with recognizing the depicted relations. Furthermore, we demonstrate that the models rely on knowledge shortcuts when answering complex diagram reasoning questions. These results suggest that the apparent successes of LVLMs in diagram reasoning tasks create a misleading impression of their true diagram understanding capabilities.

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

## A    ETHICAL CONSIDERATIONS

Our paper focuses on evaluation using synthetic and public datasets, and we thereby do not foresee any ethical issues originating from this work. The license of FoodWeb (Krishnamurthy et al., 2016) and AI2D (Kembhavi et al., 2016) is BSD-2-Clause and Apache-2.0 respectively.

**Reproducibility**   We have tried our best to ensure the reproducibility of our work. The datasets and models that we use in this paper are publicly available. We present all the details about our evaluation and data including the annotation in the Appendix. At the same time, we have uploaded our evaluation code, generated synthetic diagram data, annotated real diagram data, as well as the responses of LVLMs in supplementary files. Every stage of our work ranging from code to results is introduced thoroughly and can be easily reproduced.

## B    LIMITATIONS

Our work has two main limitations. First, we focus exclusively on diagrams that depict various entities and relationships. There may be other specialized types of diagrams that are not well-suited to this representation. We encourage future research to explore and analyze model performance on such diagrams. Second, while we demonstrate that LVLMs have limited diagram understanding capabilities and that their strong performance is largely due to knowledge shortcuts, we do not offer insights on how to address this issue. Future work could focus on developing strategies to enhance LVLMs' true diagram understanding abilities.

## C    RELATED WORK

**Evaluation of LVLMs.**   Various benchmarks have been introduced recently to evaluate the performance of multimodal models (e.g., LVLMs) focusing on different aspects like cognition and perception  (Fu et al., 2023), knowledge (Yu et al., 2023; Li et al., 2024; Yue et al., 2024), hallucination  (Li et al., 2023; Liu et al., 2023; Wang et al., 2023b) and reasoning with images (Yue et al., 2024). These benchmarks include a wide range of images, some from natural image datasets  (Li et al., 2023; Wang et al., 2023b) and others newly gathered or generated specifically for these studies (Liu et al., 2023; Fu et al., 2023; Yue et al., 2024; Zhang et al., 2024). The benchmarks are used either by employing a generative task of image captioning, where the model is prompted to describe the image  (Fu et al., 2023; Wang et al., 2023a), or by visual question answering (VQA), where the model is asked several questions related to the image (Yue et al., 2024; Lu et al., 2023; Zhang et al., 2024). These benchmarks evaluate model performances qualitatively and quantitatively to report on their strengths and weaknesses.

**Diagram Question Answering (DQA).**   Prior explorations on the DQA are diverse, focusing on various types of diagrams and questions. Some studies have addressed spatial reasoning challenges, by generating diagrams with basic shapes like in NLVR (Suhr et al., 2017) and ShapeWorld (Kuhnle & Copestake, 2017) or abstract representations of real-life scenarios (Antol et al., 2015; Zhang et al., 2016). Others have gathered text-book diagrams, such as FoodWebs (Krishnamurthy et al., 2016), AI2D (Kembhavi et al., 2016) and TQA (Kembhavi et al., 2017), containing diagrams from the grade-school science domain with more emphasis on relationships between diagram entities and complex concepts like procedures, taxonomies, and labeling of components. Although these datasets contain complex diagrams, most of the questions can be answered with commonsense knowledge, without visual reasoning. In contrast, CLEVR (Johnson et al., 2017) and IconQA (Lu et al., 2021) encompass simple diagrams but require a spatial and property-level understanding of the diagram.

With the development of recent LVLMs, a wave of explorations of (statistical) diagram understanding coming out. Masry et al. (2022); Islam et al. (2024) find that recent vision-language models cannot understand the basic elements in chart diagrams. Singh et al. (2024); Pan et al. (2024) reveal that these large models still struggle with understanding basic entities and relations of the diagram, which align with our findings on the synthetic diagrams. However, the rapid development of LVLMs shows that these large models have achieved almost perfect performance on real-world image understanding benchmarks (Li et al., 2021; Mathew et al., 2021), which align with our findings on real

diagrams. However, the discussion of the bizzared gap remains unclear. Our work lies in this place by exploring the role of knowledge and propose a hypothesis with experimental verifications.

**Capabilities and Limitations of LVLMs.** A common viewpoint on the success of LVLMs is optimistic. The great performance on various benchmarks is interpreted as the strong understanding abilities of large models on diagrams, such as identification ability (Chen et al., 2024) and reasoning ability (Lu et al., 2023). Besides diagrams, large language models are found powerful to solve various complex question-answering tasks requiring strong reasoning ablities (Qin et al., 2023; Bang et al., 2023; Fatemi et al., 2024). Large visual models are also shown to perform reasoning about images pretty well (Gupta & Kembhavi, 2023; Chen et al., 2023).

Another line of work focuses on the limitations of LVLMs. People find that large models act like stochastic parrots which only mindlessly mimic humans (i.e., memorizing pertaining corpus without any understanding) for both text reasoning and visual reasoning (Yang et al., 2024; Zecevic et al., 2023; Hu et al., 2024; Mao et al., 2023). Meanwhile, a rising number of evaluations on LVLMs exposing their limitations on more basic reasoning tasks such as counting (Fu et al., 2023) and perception (Yue et al., 2024). This line of work indicates that LVLMs perform diagram understanding via illusion, i.e., relying on the learned knowledge instead of real-time reasoning of the diagram.

## D  SUPPLEMENTARY TEST SUITE DETAILS

We provide additional details about our test suite here.

### D.1  QUESTION ANNOTATION DETAILS

For questions on synthetic diagrams, we annotate them as described in Tab. 1. Regarding real diagrams, we introduce their annotation details concerning the 6 domains respectively. For each domain, we choose a representative diagram to show how we annotate, as shown in Fig. 4.

For each question, we provide four options. We first annotate the correct option. The three incorrect options are sampled uniformly without replacement from the pool of correct answers for questions of the same type, excluding the current correct answer. Furthermore, we manually verify (and edit if necessary) the sampled negative options to ensure that they are not inadvertently correct answers. The four options are randomly shuffled before feeding into LVLMs.

#### D.1.1  ECOLOGY - FOOD WEB; FOOD CHAIN

Food web diagrams illustrate predatory relationships among animals (and between animals and plants or detritus) in the same environment (e.g. prairie, forests, sea, etc.). Any possible path from a plant to a top animal is a single food chain.

$Q_R(V|\textbf{KF}, \textbf{NR})$. This question type is consistently annotated as "Which entity is in the diagram?". For Fig. 4a, the correct option is "Sparrow".

$Q_R(V|\textbf{KF}, \textbf{NC})$. This question type is consistently annotated as "How many entities are in the diagram?". For Fig. 4a, the correct option is "10".

$Q_R(V|\textbf{KR}, \textbf{NR})$. The annotation of the question type is determined by the number of producers in the diagram: (a) If the diagram contains three or fewer producers, the question type is annotated as "Which producer is in the diagram?". (b) If the diagram includes more than three producers, the annotation changes to "Which producer is not in the diagram?". (c) Only in cases where the food web diagram contains no producers (i.e. in the detritus environment) do we annotate the question as "Which consumer is not in the diagram?". Fig. 4a contains two producers, "Grass" and "Blueberry Bush", which is applicable to category (a). The correct option we annotate is "Blueberry Bush". The (a)(b)(c) question subtypes account for $85.71\%$, $10.60\%$, and $3.68\%$ respectively. As the LVLM must comprehend the concepts of *producer* and *consumer* to correctly answer these questions, this question type is classified as requiring knowledge.

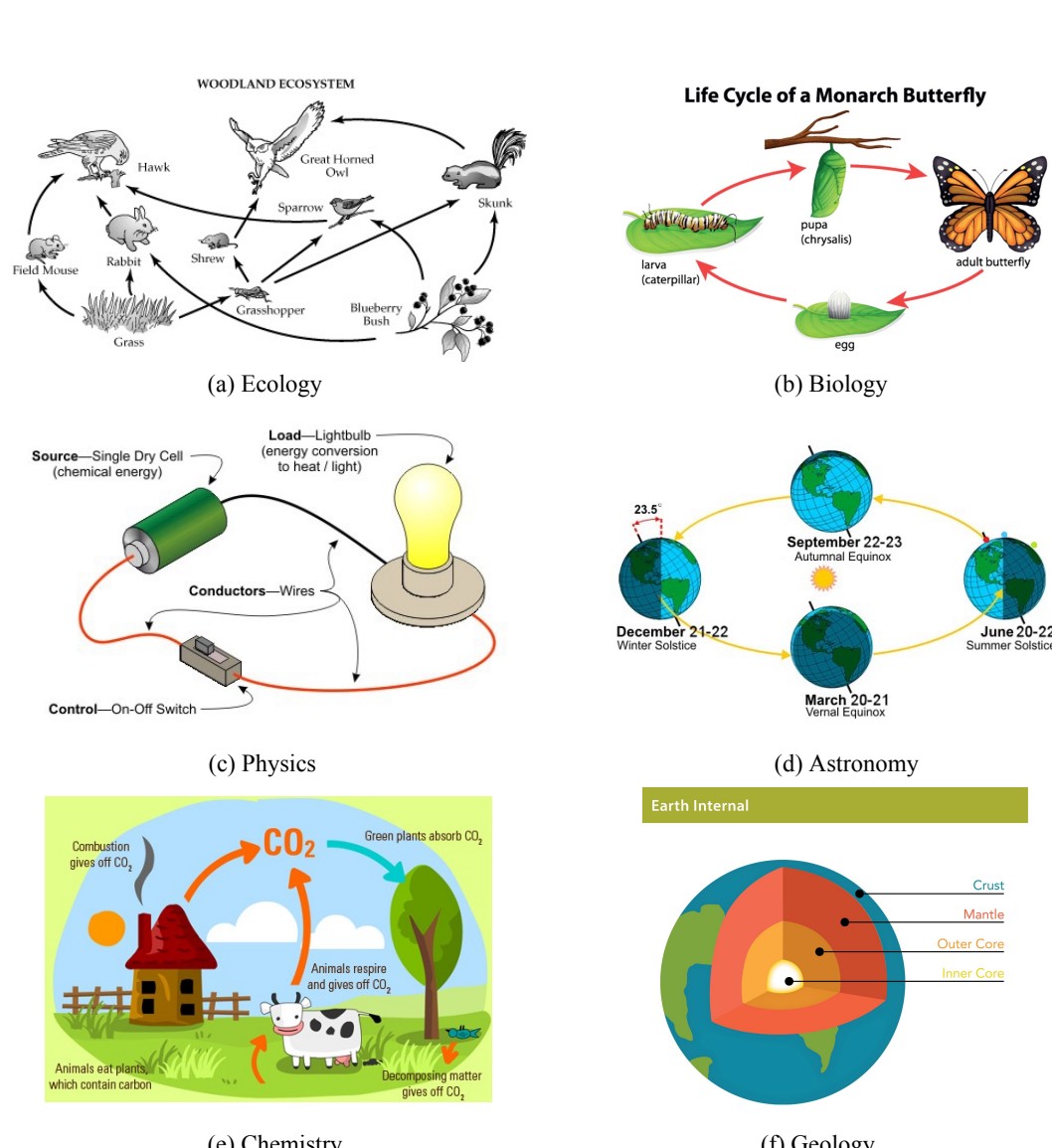

Figure 4: The representative diagrams of 6 domains.

$Q_R(V|\textbf{KR}, \textbf{NC})$. This question type is consistently annotated as "How many distinct consumers are there in the food web?", which represents a subset of $Q_R(V|\text{KF}, \text{NC})$. For Fig. 4a, the correct option is "8". Since the LVLM needs to comprehend the meaning of *consumer*, this question type is classified as requiring knowledge.

$Q_R(E|\textbf{KF}, \textbf{NR})$. Depending on the number of arrows linked to an entity, for an entity with numerous connections, the question is annotated as "Which entity is not connected to Entity?" For an entity with fewer connections, the question is annotated as "Which entity is connected to Entity?". These two subtypes account for $48.05\%$ and $51.95\%$ respectively. For Fig. 4a, we ask "Which entity is connected to Shrew", and the annotated correct option is "Great Horned Owl". This question type only involves recognizing the relations between an entity of interest and others, and thus no knowledge is required.

$Q_R(E|\textbf{KF}, \textbf{NC})$. This question type is consistently annotated as "How many arrows are linked to Entity in the diagram?". For each diagram, we randomly select an entity that has both an arrow pointing to it and an arrow pointing out of it as the entity of interest. For example, in Fig. 4a, we annotate "How many arrows are linked to Skunk in the diagram?" with the correct option "3" This question type only involves counting the relations (represented by arrows) between an entity of interest and other entities, and thus no knowledge is required.

$Q_R(E|\textbf{KR}, \textbf{NR})$. Corresponding to $Q_R(E|\text{KF}, \text{NR})$, we translate "Which entity is not connected to Entity?" to "Which is not the predator of Entity?" or "Which is not the prey of Entity?". Similarly, we translate "Which entity is connected to Entity?" to "Which is the predator of Entity?" or "Which is the prey of Entity?". Entity of interest is the same as $Q_R(E|\text{KF}, \text{NR})$. These four question subtypes account for $3.98\%$, $38.94\%$, $30.75\%$, and $26.33\%$ respectively. Refer to our annotation of Fig. 4a $Q_R(E|\text{KF}, \text{NR})$, for this question type, we annotate question as "Which is the predator of Shrew?" with the correct option "Great Horned Owl". Since the LVLM needs to comprehend the meaning of *prey* (arrow points to the entity) and *predator* (arrow points out of the entity), this question type requires knowledge.

$Q_R(E|\textbf{KR}, \textbf{NC})$. This question type is consistently annotated as "How many types of prey are consumed by Entity in the foodweb?" with the same entity of interest as $Q_R(E|\text{KR}, \text{NR})$. Accordingly, for Fig. 4a, we annotate "How many types of prey are consumed by Skunk in the foodweb?" with the correct option "2". Similarly, since the LVLM needs to comprehend the meaning of *prey*, this question type requires knowledge.

### D.1.2 BIOLOGY - LIFE CYCLE

In contrast to D.1.1, this domain delineates the developmental morphology of a single species across its life stages (e.g., for insects in diagrams like Fig. 4b, stages such as egg, pupa, and adult are depicted). These morphological stages are interconnected via arrows, forming a *directed cyclic graph*. Notably, those diagrams include arrows pointing from the mature form (adult) to its offspring (egg, embryo, or seed). We claim that the mature form represents the final stage of the species' life cycle, while the offspring represents the first stage.

$Q_R(V|\textbf{KF}, \textbf{NR})$. This question type is consistently annotated as "Which entity is in the diagram?". For Fig. 4b, the correct option is "Larva'.

$Q_R(V|\textbf{KF}, \textbf{NC})$. This question type is consistently annotated as "How many entities are in the diagram?". For Fig. 4b, the correct option is "4".

$Q_R(V|\textbf{KR}, \textbf{NR})$. This question type is annotated as "What is the [first/last] life stage for the creature in the diagram?", where we choose one of two words manually, resulting in $47.80\%$ for "first" and $52.20\%$ for "last". For Fig. 4b, we annotate "first" with the correct option "Egg". Determining the first and last stage of a lifecycle requires background knowledge in biology.

$Q_R(V|\textbf{KR}, \textbf{NC})$. This question type is consistently annotated as "How many life stages are after the stage Entity in the diagram?". For Fig. 4b, we replace Entity with "Larva" and the correct option

is "2" Since the LVLM needs to comprehend the meaning of *life stage* and the arrow direction, this question type requires knowledge.

$Q_R(E|\textbf{KF}, \textbf{NR})$. This question type is consistently annotated as "Which entity is connected to the Entity?" For Fig. 4b, we replace Entity with "Larva" and the correct option is "Egg".

$Q_R(E|\textbf{KF}, \textbf{NC})$. This question type is consistently annotated as "How many arrows are in the diagram?". In Fig. 4b, the correct option is "4".

$Q_R(E|\textbf{KR}, \textbf{NR})$. This question type is consistently annotated as "Which stage is after the Entity stage in the diagram?". For Fig. 4b, we replace Entity with "Larva" and the correct option is "Pupa". This question type requires LVLMs to understand the meaning of *stage* and arrow directions, and thus knowledge is required.

$Q_R(E|\textbf{KR}, \textbf{NC})$. This question type is consistently annotated as "How many stages can the creature change in the diagram?" Same as $Q_R(E|\text{KF}, \text{NC})$, in Fig. 4b, the correct option is "4". Similarly, since the LVLM needs to comprehend the meaning of *stage* and *creature*, this question type requires knowledge. A few diagrams use the same lifecycle paradigm to describe multiple species, e.g. egg $\rightarrow$ tadpole/chick $\rightarrow$ frog/chicken. The correct answer to $Q_R(E|\text{KR}, \text{NC})$ in this case is 3 while that to $Q_R(E|\text{KF}, \text{NC})$ is 6.

### D.1.3 Physics - Circuit

This domain contains simple middle-to-high-school circuit diagrams, typically containing only power, switches, wires, and a few appliances (e.g., light bulbs), and does not contain circuit diagrams for complex electronics that require knowledge beyond high school. Circuit diagrams can be abstracted as *undirected cyclic graphs*. Note that in this domain, we do not regard *wires* as entities even if there is explicit text in the diagram. Instead, we consider them to be the representations of relations.

$Q_R(V|\textbf{KF}, \textbf{NR})$. This question type is consistently annotated as "Which entity is in the diagram?". For Fig. 4c, the correct option is "Bulb".

$Q_R(V|\textbf{KF}, \textbf{NC})$. This question type is consistently annotated as "How many entities are in the diagram?". For Fig. 4c, the correct option is "3".

$Q_R(V|\textbf{KR}, \textbf{NR})$. This question type is annotated as "Which electronic component is in the diagram?". For Fig. 4c, the correct option is "Battery". Knowledge is required to understand the meaning of *electronic component*.

$Q_R(V|\textbf{KR}, \textbf{NC})$. This question type is consistently annotated as "How many bulbs in the diagram will glow when the switch is closed?" In Fig. 4c, the correct option is "1" Since the LVLM needs to understand the function of *switch* as well as determine whether it is a *closed circuit*, this question type is classified as requiring knowledge. All the diagrams contain switches, but some do not contain light bulbs or they are intentionally short-circuited. In such cases, the correct answer is 0.

$Q_R(E|\textbf{KF}, \textbf{NR})$. This question type is consistently annotated as "Which entity is connected to the Entity by the line?". For Fig. 4c, we replace Entity with "Bulb" and the correct option is "Battery".

$Q_R(E|\textbf{KF}, \textbf{NC})$. This question type is consistently annotated as "How many line segments are in the diagram?". For Fig. 4c, the correct option is "3".

$Q_R(E|\textbf{KR}, \textbf{NR})$. This question type is consistently annotated as "Which electronic component is connected to the Entity by the wire?" Same as $Q_R(E|\text{KF}, \text{NR})$, in Fig. 4c, we replace Entity with "Bulb"and the correct option is "Battery". This question type requires LVLMs to understand the meaning of *electronic component* and *wire*, and thus knowledge is required.

$Q_R(E|\textbf{KR}, \textsc{NC})$. This question type is consistently annotated as "How many wires are in the diagram?". Same as $Q_R(E|\textsc{KF}, \textsc{NC})$, in Fig. 4c, the correct option is "3". Similarly, since the LVLM needs to comprehend the meaning of *wire*, this question type requires knowledge.

### D.1.4 ASTRONOMY - SOLAR SYSTEM; SATELLITE SYSTEM

The subject of the diagrams in this domain encompasses seasonal changes caused by the Earth's revolution around the Sun, moon phase changes caused by the rotation of satellites around the planets, and planetary revolutions in the solar system. Diagrams describing phase changes can be regarded as *directed cyclic graphs*. A diagram depicting the solar system uses relative positions to express the relation between astronomical objects without using arrows.

$Q_R(V|\textbf{KF}, \textbf{NR})$. This question type is consistently annotated as "Which entity is in the diagram?". For Fig. 4d, the correct option is "Sun".

$Q_R(V|\textbf{KF}, \textsc{NC})$. This question type is consistently annotated as "How many entities are in the diagram?". For Fig. 4d, the correct option is "5".

$Q_R(V|\textbf{KR}, \textbf{NR})$. This question type is annotated as "Which astronomical object is in the diagram?". For Fig. 4d, the correct option is "Sun". Knowledge is required to understand the meaning of *astronomical object*.

$Q_R(V|\textbf{KR}, \textsc{NC})$. This question type is consistently annotated as "How many planets or satellites are in the diagram?" For Fig. 4d, the correct option is "4". Since the LVLM needs to understand the meaning of *planets* and *satellites*, this question type requires background knowledge.

$Q_R(E|\textbf{KF}, \textbf{NR})$. This question type is consistently annotated as "Which entity is connected to the Entity in the diagram?". For Fig. 4d, we replace Entity with "September" and the correct option is "December".

$Q_R(E|\textbf{KF}, \textsc{NC})$. This question type is consistently annotated as "How many arrows are in the diagram?". For Fig. 4d, the correct option is "5". Some of the diagrams rely on relative positions rather than arrows to represent relations between entities, in which case the answer is 0.

$Q_R(E|\textbf{KR}, \textbf{NR})$. This question type is consistently annotated as "What is the next phase after the Entity in the diagram?". For Fig. 4d, we replace Entity with "Earth in Summer" and the correct option is "Earth in Fall" This question type requires LVLMs to understand the meaning of *phase*, and thus knowledge is required.

$Q_R(E|\textbf{KR}, \textsc{NC})$. This question type is consistently annotated as "How many times that the planets or satellites can change in the diagram?". For Fig. 4d, the correct option is "4". Similarly, since the LVLM needs to comprehend the meaning of *planets* and *satellites*, this question type requires knowledge.

### D.1.5 CHEMISTRY - WATER CYCLE; CARBON CYCLE

This domain includes topics such as the water cycle and the carbon cycle where various plants and animals participate, and photosynthesis and transpiration of a specific plant. Water or carbon enjoy multiple pathways to transfer between two phases, so that substantial diagrams can be viewed as *directed multigraphs*. Other diagrams that describe a single cyclic pathway (such as photosynthesis in a single plant) can be viewed as *directed cyclic graphs*.

For example, the visual entities in the Fig. 4e are Sun, House Emissions, Carbon Dioxide, Cow, Ground (Soil), Tree, Worm, and Cloud. Among these, Smoke, Cow, Ground, Tree, and Worm are actively involved in the depicted carbon cycle. These entities are called *cycle stages*. Entities such as soil, lakes, and forests are, as per image segmentation principles, considered as single entities despite their spatial extent.

$Q_R(V|\textbf{KF}, \textbf{NR})$. This question type is consistently annotated as "Which entity is in the diagram?". For Fig. 4e, the correct option is "Cow".

$Q_R(V|\textbf{KF}, \textbf{NC})$. This question type is consistently annotated as "How many entities are in the diagram?". As mentioned in the domain summary, there are "8" visual entities in Fig. 4e.

$Q_R(V|\textbf{KR}, \textbf{NR})$. This question type is annotated as "Which cycle stage is described in the diagram?". For Fig. 4e, the correct option is "Animal".

$Q_R(V|\textbf{KR}, \textbf{NC})$. This question type is consistently annotated as "How many different cycle stages are in the diagram?" As mentioned before, there are "5" cycle stages in Fig. 4e. Since the LVLM needs to understand the function of *cycle stages*, this question type requires knowledge.

$Q_R(E|\textbf{KF}, \textbf{NR})$. This question type is consistently annotated as "Which entity is connected to the Entity by the arrow?". For Fig. 4e, we replace Entity with "Cow" and the correct option is "Carbon Dioxide"

$Q_R(E|\textbf{KF}, \textbf{NC})$. This question type is consistently annotated as "How many arrows are in the diagram?". In Fig. 4e, there are "1" arrows.

$Q_R(E|\textbf{KR}, \textbf{NR})$. This question type is consistently annotated as "Which cycle stage will happen after the Entity in the diagram?". For Fig. 4e, we replace Entity with "Animal" and the correct option is "Carbon Dioxide" This question type requires LVLMs to understand the meaning of *cycle stage*, and thus knowledge is required.

$Q_R(E|\textbf{KR}, \textbf{NC})$. This question type is consistently annotated as "How many different processes of transitions are in the diagram?" For Fig. 4c, the correct option is "5". Similarly, since the LVLM needs to comprehend the meaning of *processes of transitions*, this question type requires knowledge.

### D.1.6 GEOLOGY - PLANET STRUCTURE; STAR STRUCTURE

Diagrams in this domain depict the geological structure of the Earth or other astronomical objects, showing the relationship between geological strata through their relative positions (typically nested structures) rather than explicit arrows.

$Q_R(V|\textbf{KF}, \textbf{NR})$. This question type is consistently annotated as "Which layer is included in the diagram?". For Fig. 4f, the correct option is "Crust".

$Q_R(V|\textbf{KF}, \textbf{NC})$. This question type is consistently annotated as "How many layers are in the diagram?". For Fig. 4f, the correct option is "4".

$Q_R(V|\textbf{KR}, \textbf{NR})$. This question type is annotated as "Which stratification that is outside the core is included in the diagram?". For Fig. 4f, the correct option is 'Crust" Understanding the meaning of *stratification* and determining the containing relations between stratigraphic layers requires the use of background knowledge.

$Q_R(V|\textbf{KR}, \textbf{NC})$. This question type is consistently annotated as "How many stratifications are outside the core in the diagram?". For Fig. 4f, the correct option is "2" ("Mantle" and "Crust"). Same as above, answering this question also requires background knowledge.

$Q_R(E|\textbf{KF}, \textbf{NR})$. This question type is consistently annotated as "Which layer is next to the Layer in the diagram?". For Fig. 4f, we replace Layer with "Crust" and the correct option is "Mantle". This question type only involves recognizing the adjacency between a layer of interest and others, and thus no knowledge is required.

$Q_R(E|\textbf{KF}, \textbf{NC})$. This question type is consistently annotated as "How many layer boundaries are in the diagram?". For Fig. 4f, the correct option is "3". We regard the boundary between layers as the representation of relation. This question only involves counting the number of boundaries, so no background knowledge is required.

$Q_R(E|\textbf{KR}, \textbf{NR})$. This question type is consistently annotated as "Which stratification is the next outside layer of the Layer in the diagram?". For Fig. 4f, we replace Layer with "Mantle" and the correct option is "Crust". This question type requires LVLMs to understand the meaning of *stratification* and the containing relations between layers, hence knowledge is required.

$Q_R(E|\textbf{KR}, \textbf{NC})$. This question type is consistently annotated as "How many transition zones of the structure are in the diagram?". For Fig. 4f, the correct option is "3". Similarly, since the LVLM needs to comprehend the meaning of *transition zones*, this question type requires knowledge.

# E    SUPPLEMENTARY RESULTS

We introduce the details of the models we evaluate and provide additional results in this section.

## E.1    MODEL CONFIGURATIONS

Generally, we spend around $800\$$ for all experiments. The LVLM models are provided with the system message: "You are a visual assistant answering multiple choice questions about diagrams. Read the question, inspect the diagram, and answer with the correct choice in the following format: 'A) 0'."

**GPT-4V.** Model is used with the key `gpt-4-vision-preview` with the OpenAI API, Chat Completions. The `temperature` parameter is set to 0 to ensure deterministic outputs and a seed is given to the model to help with reproducibility. The `max_tokens` is limited to 600.

**GPT-4o.** Model is used with the key `gpt-4o` with the OpenAI API, Chat Completions. The `temperature` parameter is set to 0 to ensure deterministic outputs and a seed is given to the model to help with reproducibility. The `max_tokens` is limited to 600.

**Gemini.** Model is used with the key `gemini-1.5-pro` with the VertexAI API, Generative Models. The `temperature` parameter is set to 0 to ensure deterministic outputs. The `max_output_tokens` is limited to 600.

## E.2    SYNTHETIC DIAGRAM

The additional results on the synthetic diagrams are given in this subsection.

### E.2.1    ENTITY POSITION AND SPATIAL RELATION

| Synthetic Diagram | Question | Question Example (with Answer Options) |
|---|---|---|
| Entity Position | $Q_S(V|\text{KF}, \text{NR})$ | Which one of the text labels exists in the top row of the diagram? |
| | $Q_S(V|\text{KF}, \text{NC})$ | How many text labels are there in the top row of the diagram? |

Table 9: The template and example of questions for the evaluation of entity position and spatial relation in synthetic diagrams. The text with underline (e.g., top row) is specific and varies across diagrams.

**Preparation.** We generate synthetic diagrams similar to the previous settings (Tab. 1). We introduce a grid structure to describe the absolute positions of entities. Specifically, we use a $3 \times 3$ grid and gridlines to define the compartments in the canvas. The entities are placed in the center of them with generated arrows connecting them. We represent the entity via text. As depicted in the example in Tab. 9, we use "top/center/bottom row/column" to describe the entity location.

| Accuracy (%) | $Q_S(V|\text{KF}, \text{NR})$ | $Q_S(V|\text{KF}, \text{NC})$ |
|---|---|---|
| GPT-4V (ZS/CoT) | 75.25 / 77.67 | 41.75 / 64.02 |
| GPT-4o (ZS/CoT) | 78.07 / 89.54 | 50.30 / 79.52 |
| Gemini (ZS/CoT) | 63.58 / 64.79 | 66.80 / 73.96 |
| **Average (ZS/CoT)** | 72.30 / 77.33 | 52.95 / 72.50 |

Table 10: Performance of LVLMs on entity position QA. LVLMs can capture part of the position information and struggle with entity identification and reasoning.

**Results.** LVLMs start to struggle with identifying the entity's position attribute (Tab. 10). Even with CoT prompting, the average score of NR questions is 77.33%, which is worse compared to the entity text recognition accuracy in Tab. 3 (i.e., 95.02%). Since LVLMs only identify the position partially, the average accuracy on NC questions (i.e., 72.50% with CoT prompting) is also worse than that of text entity (i.e., 98.46%), where Gemini performs much worse than the other two models.

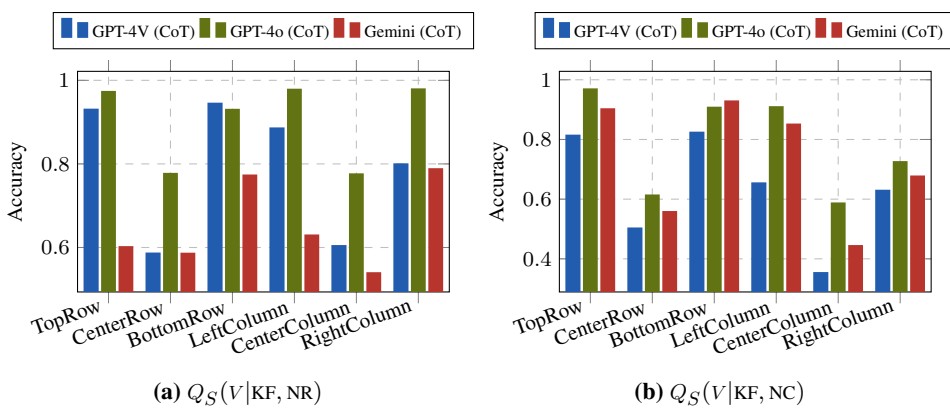

**(a)** $Q_S(V|\text{KF}, \text{NR})$      **(b)** $Q_S(V|\text{KF}, \text{NC})$

Figure 5: Accuracies of LVLMs on $Q_S(V|\text{KF}, \text{NR})$ and $Q_S(V|\text{KF}, \text{NC})$ with entities located in different positions (top row, center row, bottom row, left column, center column, and right column).

**Analysis.** We further analyze the results for more insights by visualizing the performance of LVLMs with entities in different compartments. Results (Fig. 5) show that LVLMs can answer NR and NC questions much better if the entities are not in the center area, where this phenomenon is more obvious for two GPT models.

### E.2.2 CONSISTENCY: DIAGRAM VARIATION

**Preparation.** We change the relation attributes, i.e., the arrow features in synthetic diagrams, to see if LVLMs can understand relations better. Specifically, we randomly change the arrowhead size to its 1.5, or 2 times, change the line width to its 0.5, 2, or 4 times, and the arrow color to *black*, *red*, or *blue*. Other settings remain the same as in § 2.3. Then, we ask the same questions on these newly generated diagrams to observe how performance changes. For simplicity, we only consider the CoT prompting setting since it achieves better results. The prompting templates as well as demonstration examples are shown in Figs. 21 and 22.

**Results.** We visualize the results in Fig. 6 comparing with the results in Tab. 4. We find that changing the relation attributes does not yet improve the accuracy of QA for both NR questions and NC questions. The maximum improvement is only 1%, while the average accuracies remain roughly the same (1.74% lower for NR questions and 0.3% higher for NC questions). These results further support that our findings on relations are valid and that LVLMs indeed do not understand relations even with different types of relations.

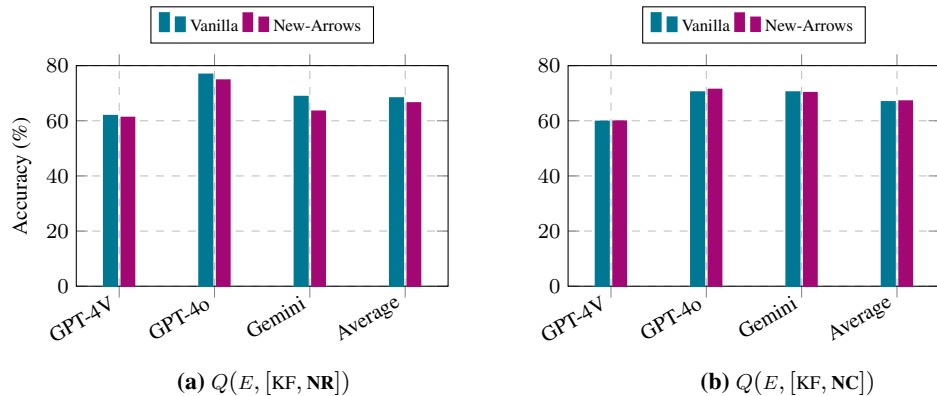

**(a)** $Q(E, [\text{KF}, \textbf{NR}])$          **(b)** $Q(E, [\text{KF}, \textbf{NC}])$

Figure 6: Accuracies of LVLMs on the diagrams with modified arrow features (denoted by New-Arrows). New results are consistent with our previous findings (denoted by Vanilla) as in § 2.3.

### E.2.3 Consistency: Prompt Variation

**Preparation.** We follow the settings in Li et al. (2024) to construct ICL prompts. We randomly select 4 examples and concatenate these diagrams as well as questions and answers as few-shot examples (represented by image). Then, we modify the template to adapt to the ICL examples under the setting of CoT prompting for evaluation. See Figs. 23 and 24 for the prompting templates and demonstration examples.

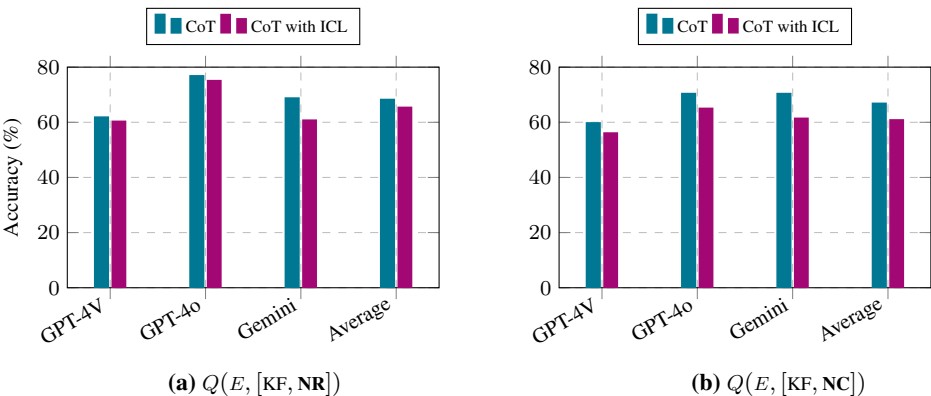

**(a)** $Q(E, [\text{KF}, \textbf{NR}])$          **(b)** $Q(E, [\text{KF}, \textbf{NC}])$

Figure 7: Accuracies of LVLMs with 4 ICL examples (with CoT prompting). Results are also consistent with our previous findings in § 2.3.

**Results.** We observe that ICL does not help with the relation identification (Fig. 7a) and reasoning (Fig. 7b) at all. Overall the average scores decrease, dropping $3\%$ for NR questions and dropping $6\%$ on NC questions. The findings further support that LVLMs can neither identify nor reason about relations even when provided a few examples with answers in context.

### E.3 Real Diagram

We provide the supplementary results for evaluations on real diagrams.

### E.3.1 Prior Knowledge Does Not Help

To test if the prior knowledge in the model affect the performance, we adjust our system prompts to clearly let the model ignore its prior knowledge when answering the questions. The original system message (i.e., prompt instructions) as well as the new one are listed below.

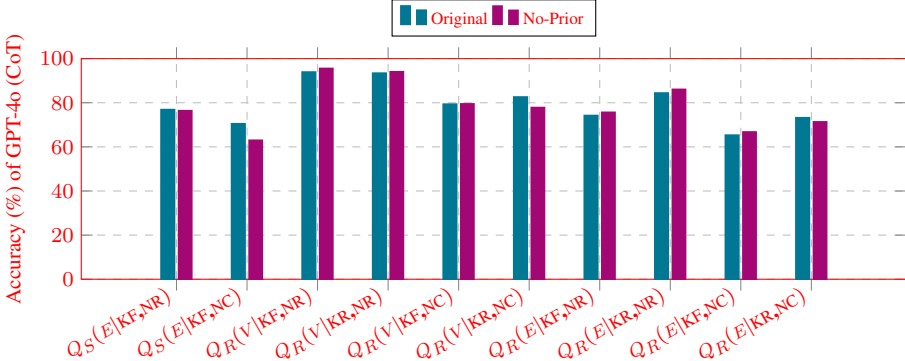

Figure 8: Test accuracies of GPT-4o under the CoT settings on both synthetic and real diagrams for the original system message (i.e., "Original") and no-prior knowledge-required system message (i.e., "No-Prior"). Results show that asking the model do not use prior knowledge could not help the model better perform the tasks.

- Original system message: "You are a visual assistant answering multiple choice questions about diagrams. Read the question, inspect the diagram, and answer with the correct choice in the following format: 'A) 0'."

- New system message that asks the model to ignore its prior knowledge: "You are a visual assistant answering multiple choice questions about diagrams. Read the question, only inspect the diagram but do not use your prior knowledge, and answer with the correct choice in the following format: 'A) 0'."

We test GPT-4o model with these two prompts under the CoT setting. Fig. 8 presents the test accuracies on both system messages are more or less the same, while our original one can achieve slightly better overall performance.

### E.3.2 DIAGRAM DISTRIBUTION ON ENTITY NUMBER

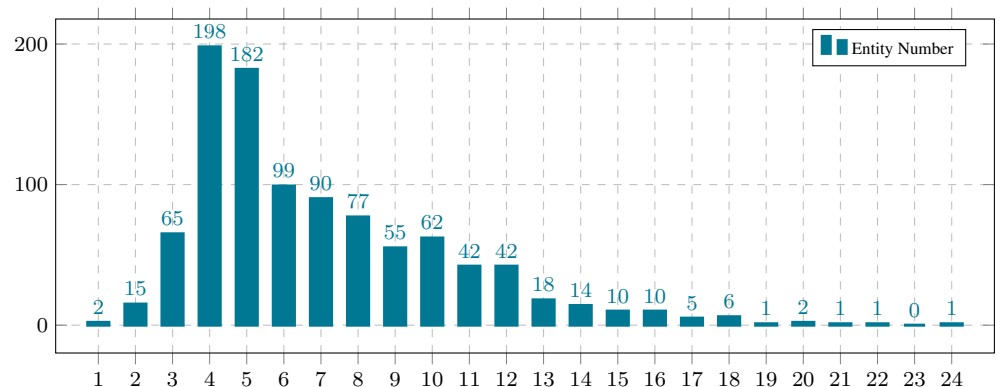

Figure 9: The distribution of the number of diagrams with respect to the number of entities in them.

For our $1,001$ real diagrams, we annotate diagrams with the number of entities in them. Thus, we visualize the distribution in Fig. 9. We can find that it is similar to the long-tail distribution, and most diagrams have $3 - 10$ entities.

### E.3.3 SUPPLEMENTARY RESULTS FOR STATISTICAL ANALYSIS

We provide all the detailed accuracies for three models on QA with respect to the number of entities (Fig. 10). Generally, all these three models have similar tendencies, and the accuracy tendencies are similar to their average as in Fig. 2.

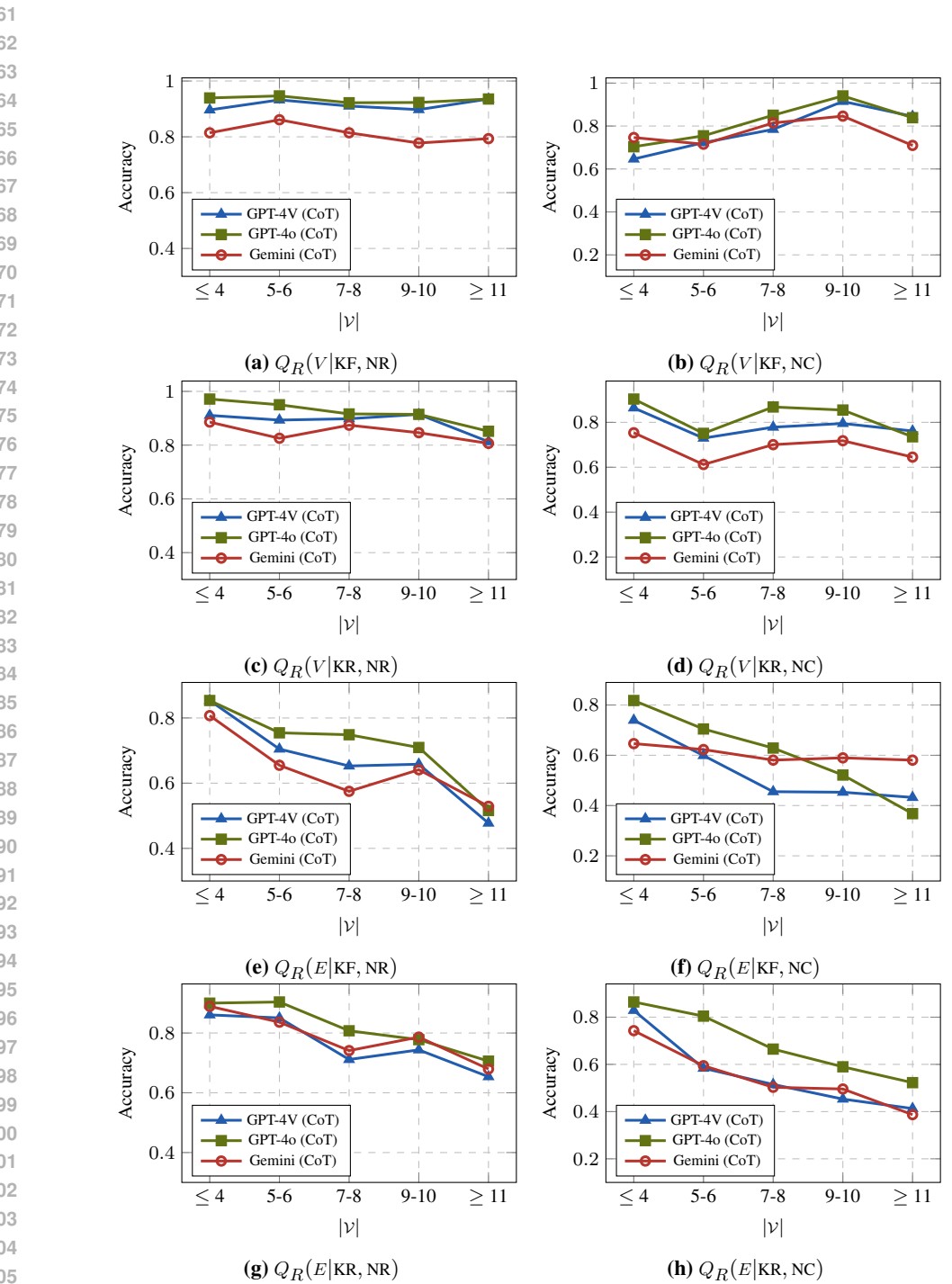

Figure 10: Accuracies of LVLMs on all questions with respect to the number of entities in the diagram.

# F EXAMPLES OF PROMPTS AND RESPONSES

## F.1 SYNTHETIC DIAGRAMS

### F.1.1 ENTITY

Prompting templates and demonstration examples for text entities and visual entities are shown in Figs. 11 to 16.

### F.1.2 RELATION

Prompting templates and demonstration examples of explicit relations $E$ (i.e., arrows) and implicit relations $E_P$ (i.e., spatial relations) are shown in Figs. 17 to 20.

### F.1.3 DIAGRAM GENERATION

Prompting templates and demonstration examples of synthetic diagrams with different arrow features are shown in Figs. 21 and 22.

### F.1.4 IN-CONTEXT LEARNING (ICL) PROMPTING TEMPLATE

Four-shot ICL Prompting templates and demonstration examples are shown in Figs. 23 and 24.

### F.1.5 DIAGRAMS WITH SEMANTIC KNOWLEDGE

Prompting templates and demonstration examples of synthetic diagrams containing semantic knowledge are shown in Figs. 25 and 26.

## F.2 REAL DIAGRAMS

Prompting templates and demonstration examples of real diagrams (Astronomy - Solar System; Satellite System) for different question types are shown in Figs. 27 to 34.

$Q_S(V|\text{KF, NR})$

**Question Prompt:** Think step by step before answering the question and show your reasoning. Which one of the entities exists in the diagram? A) bus B) hexagon C) horse **D) piggy bank**

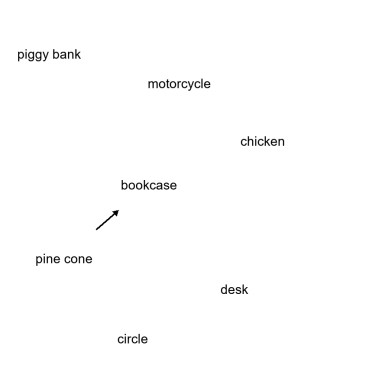

**GPT-4V:** The diagram contains labels for various objects. I will look for the entities listed in the multiple-choice options within the diagram.
A) bus - This label is not present in the diagram.
B) hexagon - There is no label for a hexagon in the diagram.
C) horse - The label "horse" is not visible in the diagram.
D) piggy bank - The label "piggy bank" is present in the top left corner of the diagram.
The correct answer is:
D) piggy bank

**GPT-4o:** To determine which entity exists in the diagram, I will examine each option and check if it is present in the diagram.
A) bus - There is no mention of a bus in the diagram.
B) hexagon - There is no mention of a hexagon in the diagram.
C) horse - There is no mention of a horse in the diagram.
D) piggy bank - The term "piggy bank" is present in the diagram.
Therefore, the correct answer is:
D) piggy bank

**Gemini 1.5 Pro:** The diagram contains the word "chicken". The answer is: C) chicken.

Figure 11: An existence question example for text entity.

$Q_S(V|\text{KF}, \text{NC})$

**Question Prompt:** Think step by step before answering the question and show your reasoning. How many text labels are there in the diagram? A) 2 **B) 3** C) 4 D) 5

**GPT-4V:** To determine the number of text labels in the diagram, I will count each instance of text:
1. "turkey"
2. "bird"
3. "chair"
There are three text labels in total.
The correct answer is: B) 3

**GPT-4o:** To determine the number of text labels in the diagram, let's count each distinct text label present:
1. "turkey"
2. "bird"
3. "chair"
There are three text labels in the diagram.
Therefore, the correct answer is: B) 3

**Gemini 1.5 Pro:** The text labels in this image are "turkey", "bird", and "chair". There are 3 text labels. **C) 3**

Figure 12: A count question example for text entity.

$Q_S(V|\text{KF}, \text{NR})$

**Question Prompt:** Think step by step before answering the question and show your reasoning. Which one of the entities exists in the diagram? A) lion B) tree house **C) fence** D) hay bale

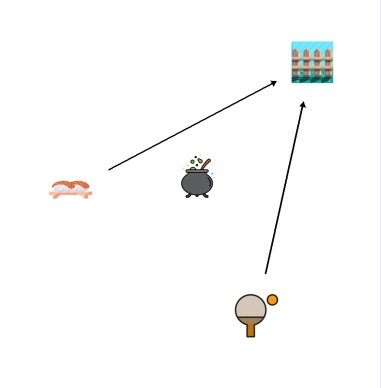

**GPT-4V:** To answer the question, I will inspect each item in the diagram and compare it to the entities listed in the options:
- A) lion: There is no image of a lion in the diagram.
- B) tree house: The image in the top right corner resembles a structure with multiple windows and a roof, which could be interpreted as a tree house.
- C) fence: There is no image of a fence in the diagram.
- D) hay bale: There is no image that resembles a hay bale in the diagram.
Based on the inspection, the entity that exists in the diagram that most closely resembles one of the options is the tree house.
B) tree house

**GPT-4o:** To determine which entity exists in the diagram, let's examine each option and compare it to the images present in the diagram:
A) Lion - There is no image of a lion in the diagram.
B) Tree house - There is no image of a tree house in the diagram.
C) Fence - There is no image of a fence in the diagram.
D) Hay bale - There is no image of a hay bale in the diagram.
The images present in the diagram are: 1. Sushi
2. A cauldron
3. A building
4. A table tennis paddle and ball
None of these images correspond to a lion, tree house, fence, or hay bale. Therefore, none of the entities listed in the options exist in the diagram.

**Gemini 1.5 Pro:** The image contains a fence. C) fence.

Figure 13: An existence question example for the visual entity.

$Q_S(V|\text{KF, NC})$

**Question Prompt:** Think step by step before answering the question and show your reasoning. How many icons are there in the diagram? A) 3 **B) 7** C) 4 D) 9

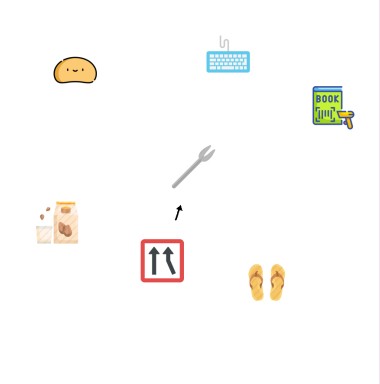

**GPT-4V:** To determine the number of icons in the diagram, I will count each distinct icon:
1. The first icon at the top left appears to be a loaf of bread.
2. The second icon at the top middle is a keyboard.
3. The third icon at the top right is a shopping cart with books.
4. The fourth icon at the bottom left is a bag of coffee beans.
5. The fifth icon at the bottom middle is a pair of tongs.
6. The sixth icon at the bottom right is a pair of flip-flops.
7. The seventh icon in the middle is a sign with arrows pointing up.
There are a total of 7 icons in the diagram.
The correct answer is: B) 7

**GPT-4o:** To determine the number of icons in the diagram, let's count each distinct icon:
1. Bread icon (top left)
2. Keyboard icon (top center)
3. Book icon (top right)
4. Milk and cookies icon (bottom left)
5. Fork icon (center)
6. Road sign icon (bottom center)
7. Sandals icon (bottom right)
There are 7 distinct icons in the diagram.
Therefore, the correct answer is: B) 7

**Gemini 1.5 Pro:** Let's count the icons in the image: bread, keyboard, book, fork, sign, sandals, milk and cookies. There are 8 icons. As 8 is not an option, the closest one is 9. D) 9

Figure 14: A count question example for the visual entity.

$Q_S(V|\text{KF, NR})$

**Question Prompt:** Think step by step before answering the question and show your reasoning. Which one of the text labels exists in the bottom row of the diagram? A) pool B) ice skate **C) juice** D) ruler

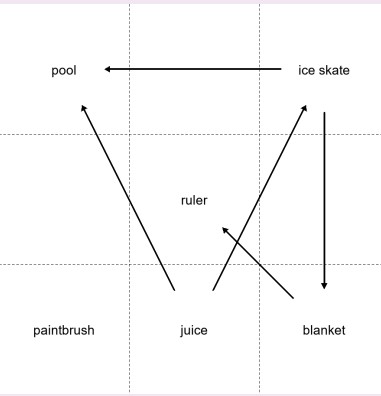

**GPT-4V:** To find the text label that exists in the bottom row of the diagram, I will look at the labels positioned at the bottom of the diagram. There are three labels in the bottom row: "paintbrush," "juice," and "blanket." Now, I will compare these labels with the options provided:
A) pool - This label is in the top row of the diagram.
B) ice skate - This label is also in the top row of the diagram.
C) juice - This label is indeed in the bottom row of the diagram.
D) ruler - This label is in the middle row of the diagram.
The correct answer is the one that matches a label in the bottom row, which is:
C) juice

**GPT-4o:** To determine which text label exists in the bottom row of the diagram, let's examine the positions of the labels:
1. The diagram is divided into a grid with three rows.
2. The bottom row contains three labels: "paintbrush," "juice," and "blanket."
Now, let's match these labels with the options provided:
A) pool - This label is in the top row.
B) ice skate - This label is in the top row.
C) juice - This label is in the bottom row.
D) ruler - This label is in the middle row.
Therefore, the correct answer is:
C) juice

**Gemini 1.5 Pro:** The text labels in the bottom row are "paintbrush", "juice", and "blanket". Of these, "blanket" is one of the choices. So the answer is D) ruler.

Figure 15: An existence question example to evaluate the position of entities.

$Q_S(V|\text{KF}, \text{NC})$

**Question Prompt:** Think step by step before answering the question and show your reasoning. How many text labels are there in the top row of the diagram? **A) 3** B) 1 C) 2 D) 0

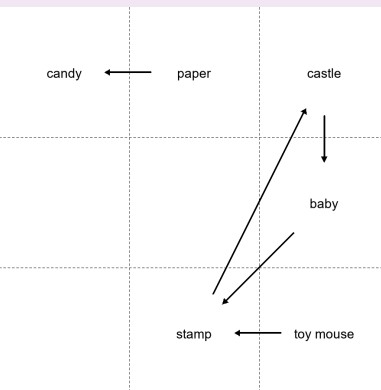

**GPT-4V:** To determine the number of text labels in the top row of the diagram, I will inspect the top row from left to right.
- The first label on the left is "candy."
- The second label in the middle is "paper."
- There is no third label in the top row.
Therefore, there are two text labels in the top row of the diagram.
C) 2

**GPT-4o:** To determine the number of text labels in the top row of the diagram, follow these steps:
1. Identify the top row of the diagram. The top row is the uppermost horizontal section of the grid.
2. Count the text labels within this top row.
In the top row, we see the following text labels:
- "candy"
- "paper"
- "castle"
There are three text labels in the top row.
Therefore, the correct answer is: A) 3

**Gemini 1.5 Pro:** The top row contains the words "candy", "paper", and "castle". There are 3 words in the top row. A) 3.

Figure 16: A count question example to evaluate the position of entities.

$Q_S(E|\text{KF}, \text{NR})$

**Question Prompt:** Think step by step before answering the question and show your reasoning. Which one of the text labels is placed on the right of the entity fishing pole? A) hummingbird B) nail **C) van** D) square

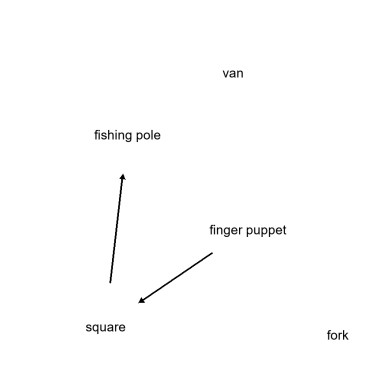

**GPT-4V:** To find the text label that is placed on the right of the entity labeled "fishing pole," I will look at the diagram and locate the "fishing pole" label. Then, I will identify which label is directly to its right.
Upon inspecting the diagram, I see that the "fishing pole" label is located towards the left side of the image. To the right of the "fishing pole" label, there is the label "van."
Therefore, the correct answer is: C) van

**GPT-4o:** To determine which text label is placed to the right of the entity "fishing pole," let's analyze the diagram step by step:
1. Locate the "fishing pole" label in the diagram. It is positioned towards the left side of the diagram.
2. Identify the labels that are to the right of the "fishing pole" label.
From the diagram: - "van" is above and to the right of "fishing pole."
- "finger puppet" is directly to the right of "fishing pole."
- "fork" is further to the right and slightly below "fishing pole."
- "square" is below and to the left of "fishing pole."
Now, let's match the options with the labels: A) hummingbird - Not present in the diagram.
B) nail - Not present in the diagram.
C) van - Present in the diagram, above and to the right of "fishing pole."
D) square - Present in the diagram, but to the left of "fishing pole."
The correct answer is: C) van

**Gemini 1.5 Pro:** The text label "finger puppet" is on the right of the entity "fishing pole". The answer is **D) square**.

Figure 17: An existence question example from the Spatial Relation subtask.

$Q_S(E|\text{KF}, \text{NC})$

**Question Prompt:** Think step by step before answering the question and show your reasoning. How many text labels are placed on the left of the entity seal? A) 2 **B) 0** C) 3 D) 1

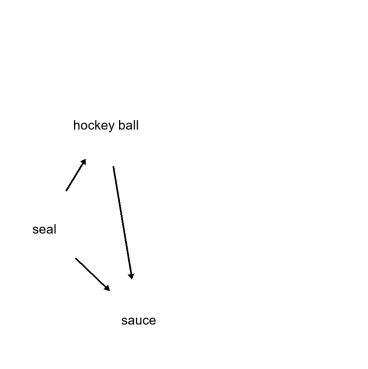

**GPT-4V:** To determine how many text labels are placed on the left of the entity "seal," I will look at the positions of the text labels relative to the word "seal" in the diagram.
1. The text label "hockey ball" is placed above and to the left of the word "seal."
2. The text label "sauce" is placed below the word "seal" and does not appear to be on the left or right, but rather directly below.
Based on the positions of the text labels, there is only one text label ("hockey ball") that is placed on the left of the entity "seal."
D) 1

**GPT-4o:** To determine how many text labels are placed on the left of the entity "seal," let's examine the diagram step by step:
1. Identify the position of the entity "seal" in the diagram.
2. Look for any text labels that are positioned to the left of "seal."
From the diagram:
- "seal" is located towards the left side of the diagram.
- The text labels "hockey ball" and "sauce" are both positioned to the right of "seal."
There are no text labels to the left of "seal."
Therefore, the correct answer is: B) 0

**Gemini 1.5 Pro:** The text label "seal" is on the left of the entity "seal". There are no other text labels on the left of "seal". Therefore, the answer is D) 1. D) 1

Figure 18: A count question example from the Spatial Relation subtask.

$Q_S(E|\text{KF, NR})$

**Question Prompt:** Think step by step before answering the question and show your reasoning. Which one of the pairs are connected in the diagram? A) gorilla and sandwich **B) penguin and sandwich** C) award ribbon and sandwich D) award ribbon and penguin

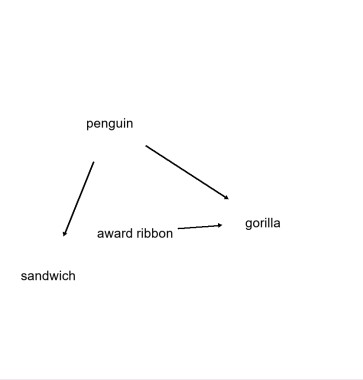

**GPT-4V:** To determine which pairs are connected in the diagram, let's look at each pair in the multiple-choice options:
A) gorilla and sandwich - There is no direct line connecting the gorilla and the sandwich in the diagram.
B) penguin and sandwich - There is no direct line connecting the penguin and the sandwich in the diagram.
C) award ribbon and sandwich - There is no direct line connecting the award ribbon and the sandwich in the diagram.
D) award ribbon and penguin - There is a direct line connecting the award ribbon and the penguin in the diagram.
Based on the connections shown in the diagram, the correct answer is:
D) award ribbon and penguin

**GPT-4o:** To determine which pairs are connected in the diagram, we need to look at the arrows and see which items are directly linked by them.
1. **Gorilla and Sandwich**: There is no direct arrow connecting "gorilla" and "sandwich".
2. **Penguin and Sandwich**: There is a direct arrow from "penguin" to "sandwich".
3. **Award Ribbon and Sandwich**: There is no direct arrow connecting "award ribbon" and "sandwich".
4. **Award Ribbon and Penguin**: There is no direct arrow connecting "award ribbon" and "penguin".
Based on the diagram, the correct answer is:
B) penguin and sandwich

**Gemini 1.5 Pro:** The award ribbon points to the gorilla and the penguin points to the gorilla. D) award ribbon and penguin

Figure 19: An existence question example from the Explicit Relation subtask.

$Q_S(E|\text{KF, NC})$

**Question Prompt:** Think step by step before answering the question and show your reasoning. How many entities are connected to ring? A) 6 B) 1 **C) 3** D) 2

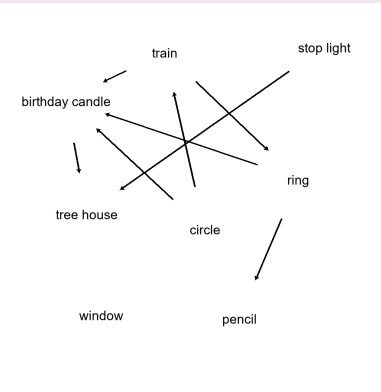

**GPT-4V:** To determine how many entities are connected to "ring," I will inspect the diagram and count the lines that connect directly to the word "ring."
Upon inspection, I can see that there are three lines connecting to "ring." These lines connect "ring" to "stop light," "circle," and "pencil."
Therefore, the correct answer is:
C) 3

**GPT-4o:** To determine how many entities are connected to "ring," we need to count the number of arrows pointing to or from "ring."
1. There is an arrow connecting "ring" to "stop light."
2. There is an arrow connecting "ring" to "circle."
3. There is an arrow connecting "ring" to "pencil."
4. There is an arrow connecting "ring" to "train."
5. There is an arrow connecting "ring" to "birthday candle."
6. There is an arrow connecting "ring" to "tree house."
So, there are 6 entities connected to "ring."
The correct answer is: A) 6

**Gemini 1.5 Pro:** Ring is connected to stop light and pencil. Therefore, there are 2 entities connected to ring. **D) 2**

Figure 20: A count question example from the Explicit Relation subtask.

$Q_S(E|\text{KF}, \text{NR})$

**Question Prompt:** Think step by step before answering the question and show your reasoning. Which one of the pairs are connected in the diagram? A) arrow and bird **B) bird and domino** C) arrow and river D) filing cabinet and river

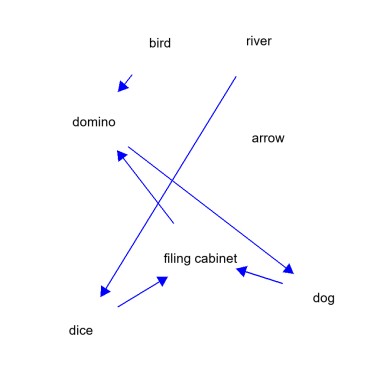

**GPT-4V:** To determine which pairs are connected in the diagram, let's look at each pair in the multiple-choice options:
A) arrow and bird - There is no line connecting the arrow to the bird.
B) bird and domino - There is no line connecting the bird to the domino.
C) arrow and river - There is a line connecting the arrow to the river.
D) filing cabinet and river - There is no line connecting the filing cabinet to the river.
Based on the connections shown in the diagram, the correct answer is:
C) arrow and river

**GPT-4o:** To determine which pairs are connected in the diagram, let's examine each pair one by one:
A) Arrow and bird: - The arrow labeled "arrow" is pointing towards the "dog". - The "bird" is connected to the "domino". - Therefore, the arrow and bird are not connected.
B) Bird and domino: - The "bird" is connected to the "domino" with a direct line. - Therefore, the bird and domino are connected.
C) Arrow and river: - The "arrow" is pointing towards the "dog". - The "river" is not connected to the "arrow". - Therefore, the arrow and river are not connected.
D) Filing cabinet and river: - The "filing cabinet" is connected to the "dog" and "dice". - The "river" is not connected to the "filing cabinet". - Therefore, the filing cabinet and river are not connected.
The correct answer is: B) bird and domino

**Gemini 1.5 Pro:** The arrow points to the dog and the filing cabinet points to the dog. The answer is D) filing cabinet and river.

Figure 21: An existence question example from the Explicit Relation subtask for a different arrow feature.

## $Q_S(E|\text{KF}, \text{NC})$

**Question Prompt:** Think step by step before answering the question and show your reasoning. How many entities are connected to apricot? A) 4 B) 0 **C) 3** D) 2

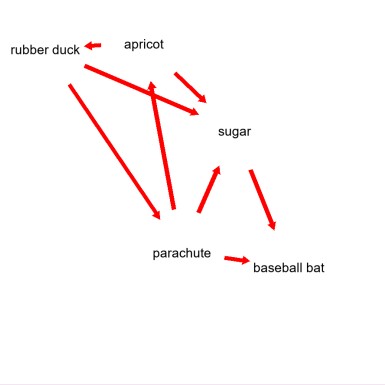

**GPT-4V:** To determine how many entities are connected to "apricot," I will count the lines that connect directly to the word "apricot" in the diagram.
1. There is one line connecting "apricot" to "rubber duck."
2. There is another line connecting "apricot" to "sugar."
3. There is a third line connecting "apricot" to "parachute."
No other lines are connected to "apricot." Therefore, there are three entities connected to "apricot."
C) 3

**GPT-4o:** To determine how many entities are connected to "apricot," we need to count the number of arrows pointing to and from "apricot."
1. There is an arrow from "apricot" to "rubber duck."
2. There is an arrow from "apricot" to "sugar."
3. There is an arrow from "sugar" to "apricot."
4. There is an arrow from "parachute" to "apricot."
So, "apricot" is connected to four entities: "rubber duck," "sugar," "sugar" (again, but counted once), and "parachute."
Therefore, the correct answer is: A) 4

**Gemini 1.5 Pro:** Apricot has connections to rubber duck, parachute, and sugar. There are 3 connections in total. Therefore, the answer is **C) 3**.

Figure 22: A count question example from the Explicit Relation subtask for a different arrow feature.

$Q_S(E|\text{KF, NR})$

**Question Prompt:** Following the in-context examples in the first image, answer the following question about the second image. Think step by step before answering the question and show your reasoning. Which one of the pairs are connected in the diagram? A) clock tower and flower B) swim fin and flower **C) clock tower and piano** D) flower and piano

In-Context Reference Examples

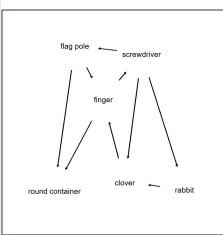 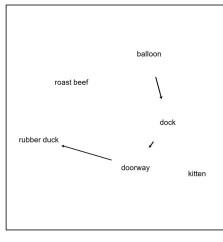 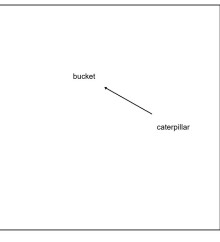 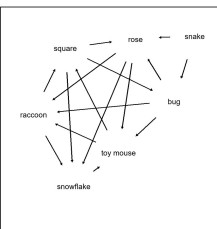

Question: How many entities are connected to finger?
A) 1
B) 2
C) 3
D) 4
Answer: D) 4

Question: Which one of the pairs are connected in the diagram?
A) clock and kitten
B) clock and doorway
C) balloon and kitten
D) balloon and doorway
Answer: B) clock and doorway

Question: How many entities are connected to caterpillar?
A) 0
B) 1
C) 2
D) 3
Answer: B) 1

Question: Which one of the pairs are connected in the diagram?
A) snake and square
B) snowflake and bug
C) rose and bug
D) snake and raccoon
Answer: C) rose and bug

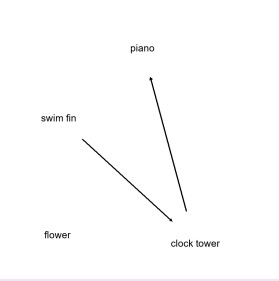

**GPT-4V:** To answer the question, I will examine the second image to determine which pairs of items are connected by lines.
- The "clock tower" is connected to the "piano" by a line. ...
Based on these observations, the correct answer is:
B) swim fin and flower

**GPT-4o:** To determine which pairs are connected in the diagram, let's examine the connections:
1. **Clock tower**: - Connected to **piano**. ...
Since we need to find the correct pair, the answer is:
B) Swim fin and flower

**Gemini 1.5 Pro:** The swim fin and clock tower are connected by a line. The answer is **C) clock tower and piano**.

Figure 23: An existence question example from the Explicit Relation subtask with few-shot examples.

$Q_S(E|\text{KF, NC})$

**Question Prompt:** Following the in-context examples in the first image, answer the following question about the second image. Think step by step before answering the question and show your reasoning. How many entities are connected to rectangle? A) 0 B) 3 **C) 2** D) 1

In-Context Reference Examples

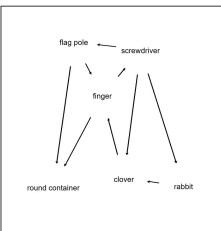 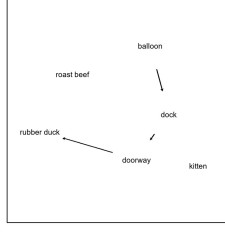 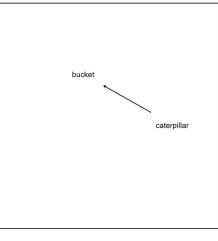 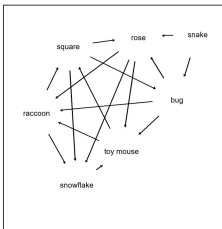

Question: How many entities are connected to finger?
A)  1
B)  2
C)  3
D)  4
Answer: D) 4

Question: Which one of the pairs are connected in the diagram?
A)  clock and kitten
B)  clock and doorway
C)  balloon and kitten
D)  balloon and doorway
Answer: B) clock and doorway

Question: How many entities are connected to caterpillar?
A)  0
B)  1
C)  2
D)  3
Answer: B) 1

Question: Which one of the pairs are connected in the diagram?
A)  snake and square
B)  snowflake and bug
C)  rose and bug
D)  snake and raccoon
Answer: C) rose and bug

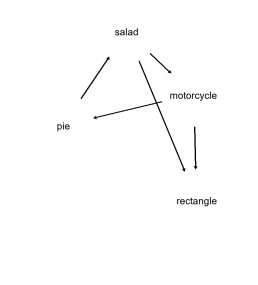

**GPT-4V:** In the second image, the entity labeled "rectangle" is connected to two other entities: "pie" and "motorcycle." Therefore, the correct answer to the question is:
C) 2

**GPT-4o:** To determine how many entities are connected to the rectangle in the second image, follow these steps:
1. Identify the rectangle in the diagram.
2. Count the number of arrows pointing to or from the rectangle.
Upon inspection: - The rectangle has two arrows pointing to it from "motorcycle."
Therefore, the number of entities connected to the rectangle is 2.
The correct answer is: C) 2

**Gemini 1.5 Pro:** The entities connected to rectangle are motorcycle and salad. Motorcycle is connected with a single line, and salad is connected with a line that goes through motorcycle. Therefore, there are **2** entities connected to rectangle. **C) 2**

Figure 24: A count question example from the Explicit Relation subtask with few-shot examples.

$Q_S(E|\text{KR, NR})$

**Question Prompt:** Think step by step before answering the question and show your reasoning. Which one of the pairs are connected in the diagram? A) butterfly and skunk B) bug and skunk C) snake and goldfish **D) caterpillar and snake**

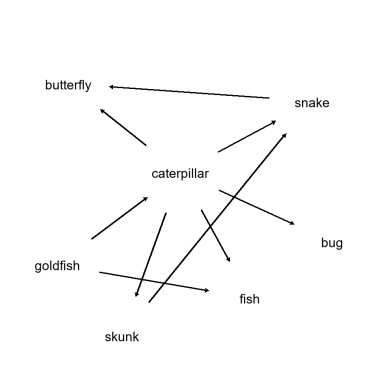

**GPT-4V:** To determine which pair of words are connected in the diagram, I will look at each pair in the multiple-choice options and see if there is a line connecting them in the diagram.
A) butterfly and skunk - There is no line directly connecting 'butterfly' and 'skunk' in the diagram.
B) bug and skunk - There is no line directly connecting 'bug' and 'skunk' in the diagram.
C) snake and goldfish - There is no line directly connecting 'snake' and 'goldfish' in the diagram.
D) caterpillar and snake - There is a line directly connecting 'caterpillar' and 'snake' in the diagram.
The correct answer is:
D) caterpillar and snake

**GPT-4o:** To determine which pair is connected in the diagram, we need to follow the arrows between the pairs mentioned in the options.
A) Butterfly and skunk: - There is no direct arrow connecting butterfly and skunk.
B) Bug and skunk: - There is no direct arrow connecting bug and skunk.
C) Snake and goldfish: - There is no direct arrow connecting snake and goldfish.
D) Caterpillar and snake: - There is a direct arrow from caterpillar to snake.
Therefore, the correct answer is: D) caterpillar and snake

**Gemini 1.5 Pro:** The caterpillar has an arrow pointing to the butterfly. The butterfly has an arrow pointing to the snake. This means the caterpillar and snake are connected. D) caterpillar and snake **D) caterpillar and snake**

Figure 25: An existence question example for a diagram generated from a knowledge graph.

$Q_S(E|\text{KR}, \text{NC})$

**Question Prompt:** Think step by step before answering the question and show your reasoning. How many entities are connected to doll? A) 2 B) 0 **C) 1** D) 3

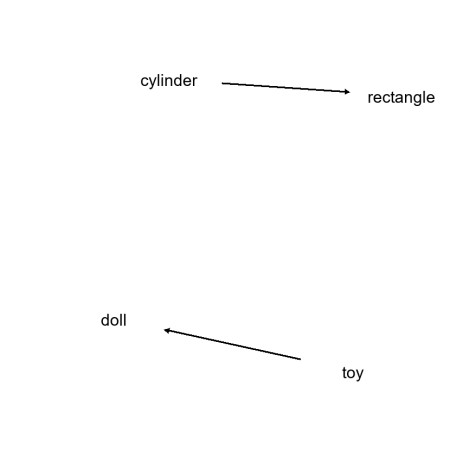

**GPT-4V:** To determine how many entities are connected to "doll," let's look at the diagram:
- There is one line connecting "doll" to "toy."
There are no other lines connecting "doll" to any other entities in the diagram. Therefore, "doll" is connected to only one entity.
The correct answer is: C) 1

**GPT-4o:** To determine how many entities are connected to "doll," let's examine the diagram step by step:
1. Identify the entities in the diagram: "cylinder," "rectangle," "doll," and "toy." 2. Look for arrows pointing to or from "doll."
From the diagram: - There is one arrow pointing from "toy" to "doll."
Therefore, "doll" is connected to one entity.
The correct answer is: C) 1

**Gemini 1.5 Pro:** Doll is connected to toy. There is an arrow pointing from toy to doll. Therefore, the answer is C) 1.

Figure 26: A count question example for a diagram generated from a knowledge graph.

$Q_R(V|\text{KR, NR})$

**Question Prompt:** Think step by step before answering the question and show your reasoning. Which astronomical object is in the diagram? **A) earth** B) moon C) belt D) jupiter

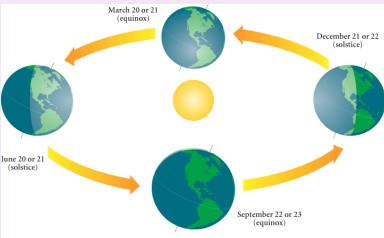

**GPT-4V:** The diagram shows a central yellow sphere representing the Sun and a blue-green sphere in four different positions around it, representing the Earth at various points in its orbit. The diagram includes labels for the equinoxes and solstices, which are events that occur due to the Earth's tilt and orbit around the Sun. The Earth is depicted with a tilt, and the different positions show how the sunlight hits the Earth differently throughout the year, leading to the changing seasons.
Given the options provided and the context of the diagram...
The correct answer is:
A) earth

**GPT-4o:** To determine which astronomical object is in the diagram, let's analyze the elements present:
1. The diagram shows a central yellow object, which is likely the Sun. 2. Surrounding the Sun, there are four blue and green spheres with landmasses that resemble continents on Earth. 3. The diagram labels specific dates: March 20 or 21 (equinox), June 20 or 21 (solstice), September 22 or 23 (equinox), and December 21 or 22 (solstice). These dates correspond to significant points in Earth's orbit around the Sun, marking the equinoxes and solstices.
Given these observations, the object in the diagram is Earth.
Therefore, the correct answer is: A) earth

**Gemini 1.5 Pro:** The diagram shows the earth revolving around the sun. So the answer is A)

Figure 27: A knowledge-required entity recognition question example for a real diagram.

2484
2485
2486
2487
2488
2489
2490
2491
2492
2493
2494
2495
2496
2497
2498
2499
2500
2501
2502
2503
2504
2505
2506
2507
2508
2509
2510
2511
2512
2513
2514
2515
2516
2517
2518
2519
2520
2521
2522
2523
2524
2525
2526
2527
2528
2529
2530
2531
2532
2533
2534
2535
2536
2537

$Q_R(V|\text{KR}, \text{NC})$

**Question Prompt:** Think step by step before answering the question and show your reasoning. How many planets or satellites are in the diagram? A) 3 B) 8 C) 10 **D) 4**

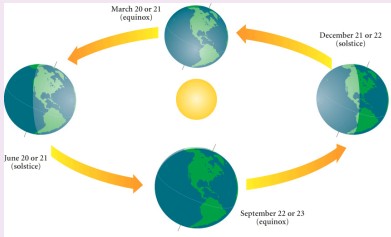

**GPT-4V:** The diagram shows four distinct celestial bodies: one is the Sun, and the other three are representations of Earth at different times of the year, indicating the position of Earth relative to the Sun during the equinoxes and solstices. These are not three different planets or satellites, but rather one planet, Earth, shown at three different points in its orbit around the Sun. Therefore, the answer to how many planets or satellites are in the diagram is:
D) 4

**GPT-4o:** To determine the number of planets or satellites in the diagram, let's carefully analyze the image step by step:
1. **Identify the objects in the diagram:** - There is a central yellow object, which represents the Sun. - There are four blue and green objects surrounding the Sun, which represent the Earth at different positions in its orbit around the Sun.
2. **Count the distinct objects:** - The Sun is one object. - The Earth is shown in four different positions, but it is still the same planet.
3. **Summarize the count:** - Total number of distinct objects: 1 (Sun) + 1 (Earth) = 2.
Since the question asks for the number of planets or satellites, and we only have the Earth (a planet) and the Sun (which is a star, not a planet or satellite), the correct count is 1 planet.
Therefore, the correct answer is: D) 4

**Gemini 1.5 Pro:** There are four images of the Earth and one of the Sun. The Earth is a planet and the Sun is a star. A) 4

Figure 28: A knowledge-required entity reasoning question example for a real diagram.

2538
2539
2540
2541
2542
2543
2544
2545
2546
2547
2548
2549
2550
2551
2552
2553
2554
2555
2556
2557
2558
2559
2560
2561
2562
2563
2564
2565
2566
2567
2568
2569
2570
2571
2572
2573
2574
2575
2576
2577
2578
2579
2580
2581
2582
2583
2584
2585
2586
2587
2588
2589
2590
2591

$Q_R(E|\text{KR}, \text{NR})$

**Question Prompt:** Think step by step before answering the question and show your reasoning. What is the next stage after the earth in summer in the diagram? A) full moon B) perigee **C) earth in fall** D) earth in june

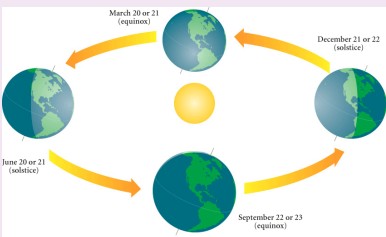

**GPT-4V:** The diagram shows the Earth at different points in its orbit around the Sun, corresponding to the seasons. The Earth in summer is labeled "June 20 or 21 (solstice)." Following the direction of the arrows, the next stage after the Earth in summer would be the Earth in fall. The Earth in fall is labeled "September 22 or 23 (equinox)."
Therefore, the correct answer is:
C) earth in fall

**GPT-4o:** To determine the next stage after the Earth in summer in the diagram, let's follow these steps:
1. Identify the current position of the Earth in summer. The diagram shows that the Earth in summer is labeled as "June 20 or 21 (solstice)". 2. Follow the direction of the arrows to see the sequence of the Earth's positions around the Sun. 3. The arrow from "June 20 or 21 (solstice)" points to "September 22 or 23 (equinox)".
Therefore, the next stage after the Earth in summer is "earth in fall".
The correct answer is: C) earth in fall

**Gemini 1.5 Pro:** The diagram shows the earth at different points in its orbit around the sun. The earth in summer (June 20 or 21) is followed by the earth in fall (September 22 or 23). **C) earth in fall**

Figure 29: A knowledge-free relation recognition question example for a real diagram.

### $Q_R(E|\text{KR, NC})$

**Question Prompt:** Think step by step before answering the question and show your reasoning. How many times that the planets or satellites can change in the diagram? A) 0 B) 12 **C) 4** D) 8

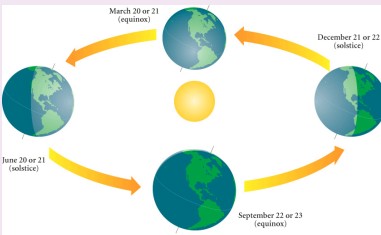

**GPT-4V:** The diagram shows the Earth at four different points in its orbit around the Sun, corresponding to the equinoxes and solstices. These are the positions where the tilt of Earth's axis is oriented such that the sun's rays are directly over the equator (equinoxes) or over the Tropic of Cancer or Capricorn (solstices), leading to the start of the different seasons.

The question seems to be asking how many times the planets or satellites can change positions in the diagram. Since the diagram is specifically showing Earth's position at four distinct times of the year, the planets or satellites (in this case, Earth) change positions four times as it moves through its orbit from one equinox or solstice to the next.

Therefore, the answer is: C) 4

**GPT-4o:** To determine how many times the planets or satellites can change in the diagram, let's analyze the diagram step by step:

1. The diagram shows the Earth at four different positions in its orbit around the Sun, corresponding to the solstices and equinoxes. 2. The positions are: - March 20 or 21 (equinox) - June 20 or 21 (solstice) - September 22 or 23 (equinox) - December 21 or 22 (solstice) 3. The arrows indicate the transition from one position to the next.

Each transition from one position to the next represents a change. There are four positions, and each position transitions to the next:

- March to June - June to September - September to December - December to March There are 4 transitions in total.

Therefore, the number of times the planets or satellites can change in the diagram is: C) 4

**Gemini 1.5 Pro:** The diagram shows the Earth at four different points in its orbit around the sun. Therefore, the planets or satellites in the diagram can change 4 times. **C) 4**

Figure 30: A knowledge-required relation reasoning question example for a real diagram.

2646
2647
2648
2649
2650
2651
2652
2653
2654
2655
2656
2657
2658
2659
2660
2661
2662
2663
2664
2665
2666
2667
2668
2669
2670
2671
2672
2673
2674
2675
2676
2677
2678
2679
2680
2681
2682
2683
2684
2685
2686
2687
2688
2689
2690
2691
2692
2693
2694
2695
2696
2697
2698
2699

$Q_R(V|\text{KF}, \text{NR})$

**Question Prompt:** Think step by step before answering the question and show your reasoning. Which entity is in the diagram? **A) earth** B) belt C) jupiter D) moon

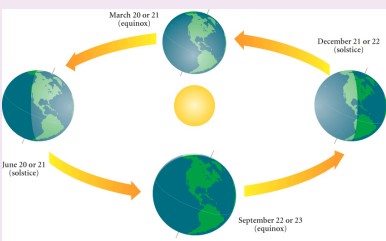

**GPT-4V:** The diagram shows a central yellow sphere representing the Sun and four different positions of a blue and green sphere representing a planet as it orbits around the Sun. The four positions correspond to different dates and are labeled with specific astronomical events: two equinoxes (March 20 or 21 and September 22 or 23) and two solstices (June 20 or 21 and December 21 or 22)...
Given the options provided:
A) Earth - This is the correct answer because the diagram depicts Earth's orbit around the Sun and the changing seasons, which are specific to Earth.
B) Belt - This is not correct; there is no belt depicted in the diagram.
C) Jupiter - This is not correct; the diagram does not represent Jupiter's orbit or its relation to the Sun.
D) Moon - This is not correct; the diagram does not show the Moon or its phases.
The correct answer is:
A) Earth

**GPT-4o:** To determine which entity is in the diagram, let's analyze the elements present:
1. The diagram shows a central yellow circle, which represents the Sun. 2. Surrounding the Sun, there are four blue and green spheres with landmasses that resemble Earth's continents. 3. The labels indicate specific dates related to equinoxes and solstices, which are events that occur due to Earth's orbit around the Sun.
Given these observations, the entity in the diagram is:
A) earth

**Gemini 1.5 Pro:** The diagram shows the earth at different points in its orbit around the sun. **A)**

Figure 31: A knowledge-free entity recognition question example for a real diagram.

$Q_R(V|\text{KF}, \text{NC})$

**Question Prompt:** Think step by step before answering the question and show your reasoning. How many entities are in the diagram? A) 15 **B) 5** C) 12 D) 2

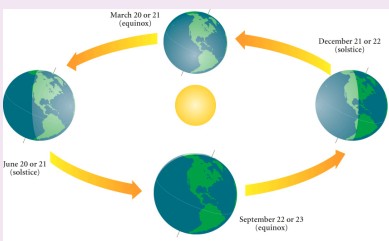

**GPT-4V:** To determine the number of entities in the diagram, let's count each distinct item:
1. The Sun (in the center) 2. Earth at the top (March equinox) 3. Earth on the right (December solstice) 4. Earth at the bottom (September equinox) 5. Earth on the left (June solstice)
Adding these up, we have a total of 5 entities.
The correct answer is: B) 5

**GPT-4o:** To determine the number of entities in the diagram, let's identify and count each distinct object:
1. The Sun in the center. 2. Four Earth positions around the Sun, each representing different times of the year (March equinox, June solstice, September equinox, December solstice).
So, we have: - 1 Sun - 4 Earth positions
Adding these together, we get a total of 5 entities.
Therefore, the correct answer is: B) 5

**Gemini 1.5 Pro:** There are 4 Earth illustrations and 1 Sun illustration. 4 + 1 = 5 **B) 5**

Figure 32: A knowledge-free entity reasoning question example for a real diagram.

$Q_R(E|\text{KF}, \text{NR})$

**Question Prompt:** Think step by step before answering the question and show your reasoning. Which entity is connected to the june in the diagram? A) waning crescent B) summer C) waxing crescent **D) september**

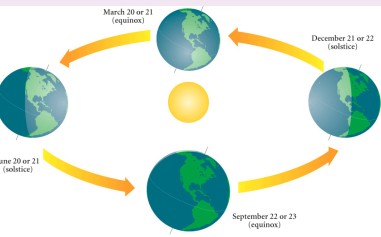

**GPT-4V:** The diagram shows the Earth at four key points in its orbit around the Sun, corresponding to the equinoxes and solstices. The date "June 20 or 21" is labeled with "(solstice)," which refers to the summer solstice in the Northern Hemisphere. This is the time when the Northern Hemisphere is tilted most directly towards the Sun and experiences the longest day of the year, marking the beginning of summer. Therefore, the entity connected to June in the diagram is:
B) summer

**GPT-4o:** To determine which entity is connected to June in the diagram, let's analyze the diagram step by step:
1. The diagram shows the Earth's position relative to the Sun at different times of the year.
2. The positions are labeled with specific dates and events...
3. The question asks for the entity connected to June. In the diagram, June 20 or 21 is labeled as a solstice.
4. The solstice in June is known as the summer solstice in the Northern Hemisphere, marking the beginning of summer.
Now, let's match this information with the given options:
A) waning crescent - This is a phase of the moon, not related to the solstice.
B) summer - This is correct, as the June solstice marks the beginning of summer in the Northern Hemisphere.
C) waxing crescent - This is a phase of the moon, not related to the solstice.
D) September - This is another month, not directly connected to June.
Therefore, the correct answer is: B) summer

**Gemini 1.5 Pro:** The arrow points from June to September. **D) september**

Figure 33: A knowledge-free relation recognition question example for a real diagram.

2808
2809
2810
2811
2812
2813
2814
2815
2816
2817
2818
2819
2820
2821
2822
2823
2824
2825
2826
2827
2828
2829
2830
2831
2832
2833
2834
2835
2836
2837
2838
2839
2840
2841
2842
2843
2844
2845
2846
2847
2848
2849
2850
2851
2852
2853
2854
2855
2856
2857
2858
2859
2860
2861

## $Q_R(E|\text{KF}, \text{NC})$

**Question Prompt:** Think step by step before answering the question and show your reasoning. How many arrows are in the diagram? **A) 4** B) 1 C) 13 D) 5

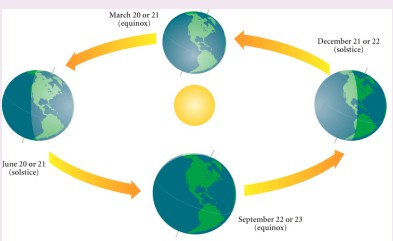

**GPT-4V:** To determine the number of arrows in the diagram, I will count each arrow that is depicted.

Starting from the top right and moving clockwise: 1. There is one large arrow pointing from the globe labeled "December 21 or 22 (solstice)" to the globe labeled "March 20 or 21 (equinox)". 2. Another large arrow points from the globe labeled "March 20 or 21 (equinox)" to the globe labeled "June 20 or 21 (solstice)". 3. A third large arrow points from the globe labeled "June 20 or 21 (solstice)" to the globe labeled "September 22 or 23 (equinox)". 4. Finally, a fourth large arrow points from the globe labeled "September 22 or 23 (equinox)" back to the globe labeled "December 21 or 22 (solstice)".

There are no other arrows visible in the diagram. Therefore, the total count is four arrows.

The correct answer is: A) 4

**GPT-4o:** To determine the number of arrows in the diagram, let's carefully count each arrow step by step:

1. Starting from the top Earth (March 20 or 21), there is an arrow pointing to the right towards the Earth labeled December 21 or 22. 2. From the Earth labeled December 21 or 22, there is an arrow pointing downwards towards the Earth labeled September 22 or 23. 3. From the Earth labeled September 22 or 23, there is an arrow pointing to the left towards the Earth labeled June 20 or 21. 4. From the Earth labeled June 20 or 21, there is an arrow pointing upwards towards the Earth labeled March 20 or 21.

So, there are a total of 4 arrows in the diagram.

The correct answer is: A) 4

**Gemini 1.5 Pro:** There is one arrow connecting each globe. There are four globes. Therefore, there are four arrows. **A) 4**

Figure 34: A knowledge-free relation reasoning question example for a real diagram.

