# OpenReview forum: "Do Vision-Language Models Really Understand Visual Language?"
_ICLR.cc/2025/Conference — Submitted to ICLR 2025_

### Official Review · Reviewer_TikE · 2024-11-02

**Soundness:** 3
**Presentation:** 3
**Contribution:** 2
**Rating:** 5
**Confidence:** 2

**Summary:**

This paper proposes an evaluation framework to assess LVLMs' capability in understanding diagrams. The results show that models like GPT-4V and Gemini rely primarily on their knowledge base rather than reasoning abilities when understanding relationships.
However, contributions of this paper are not particularly prominent. Is the core contribution the proposed test suite, the evaluation dataset, or the insights gained? In fact, some of these insights have already been revealed, such as “models consistently perform well on entity-related questions”, “models exhibit significant difficulty in identifying relationships”. I suggest that authors review the existing work and clearly highlight the differences between this paper and previous work, including the evaluation methods and insights gained.
Additionally, beyond the overall evaluation results, I would like to see specific results across different domain and topics (as shown in Table 2). This could lead to new discoveries and spark more valuable discussions (involving different knowledge systems).

The Chain of Thought approach mentioned by the authors is relatively simple, as it only involves adding prompts like “think Step-by-Step.” Are there any more effective ways to use Step-by-Step prompts for this kind of chart? I suggest that the authors explore this in greater depth. In the authors’ experiments, it is mentioned that under certain circumstance, the test of LVLMs do not use any knowledge shortcuts. However, for some simple relational charts, how can we be certain that these large models have not encountered similar charts during training? This point remains uncertain.

By the way, in this work, beyond the evaluations conducted by the authors, I would like to see a model proposed by the authors specifically for understanding basic relational charts.

**Strengths:**

The overall contribution of the paper lies in the comprehensive nature of the experiments, including a relatively thorough consideration of chart understanding (both explicit and implicit) and other relevant aspects.

**Weaknesses:**

However, contributions of this paper are not particularly prominent. Is the core contribution the proposed test suite, the evaluation dataset, or the insights gained? In fact, some of these insights have already been revealed, such as “models consistently perform well on entity-related questions”, “models exhibit significant difficulty in identifying relationships”. I suggest that authors review the existing work and clearly highlight the differences between this paper and previous work, including the evaluation methods and insights gained.

**Questions:**

Additionally, beyond the overall evaluation results, I would like to see specific results across different domain and topics (as shown in Table 2). This could lead to new discoveries and spark more valuable discussions (involving different knowledge systems).

The Chain of Thought approach mentioned by the authors is relatively simple, as it only involves adding prompts like “think Step-by-Step.” Are there any more effective ways to use Step-by-Step prompts for this kind of chart? I suggest that the authors explore this in greater depth. In the authors’ experiments, it is mentioned that under certain circumstance, the test of LVLMs do not use any knowledge shortcuts. However, for some simple relational charts, how can we be certain that these large models have not encountered similar charts during training? This point remains uncertain.

By the way, in this work, beyond the evaluations conducted by the authors, I would like to see a model proposed by the authors specifically for understanding basic relational charts.

---

> ### Author Response · Authors · 2024-11-13
> **Response to Review TikE**
>
> We thank the reviewer for the time! We would kindly point out that there are some misunderstandings about our work, and we would like to respond point by point for elaboration.
>
>
> >***Point 1**: However, contributions of this paper are not particularly prominent. Is the core contribution the proposed test suite, the evaluation dataset, or the insights gained?*
>
> >**Response**: Our core contribution is not only the test suite or datasets but the resulting insights which may drive the future design of datasets and models. We mentioned our contribution in the abstract, introduction, as well as contribution sections. For example in the conclusion section, we mention:
> >
> >>We evaluate three LVLMs on diagram understanding, and find that while these models can perfectly recognize and reason about entities depicted in the diagrams, they struggle with recognizing the depicted relations. Furthermore, we demonstrate that the models primarily rely on knowledge shortcuts when answering complex diagram reasoning questions. These results suggest that the apparent successes of LVLMs in diagram reasoning tasks create a misleading impression of their true diagram understanding capabilities.
> >
> >To do that, we proposed a test suite, generating and annotating new datasets. These contributions are also helpful for the community and future explorations.
>
> ---
>
> >***Point 2**: In fact, some of these insights have already been revealed, such as “models consistently perform well on entity-related questions”, “models exhibit significant difficulty in identifying relationships”. I suggest that authors review the existing work and clearly highlight the differences between this paper and previous work, including the evaluation methods and insights gained.*
>
> >**Response**: According to our knowledge, our work first obtains these insights for large models on knowledge-rich diagrams. As discussed with Reviewer 6WSg, we include more discussions about other diagrams without rich background knowledge such as charts and graphs in the related work (see red lines 912-920). We would appreciate it if the reviewer could provide more specific references to help us further improve the related work section.
>
> ---
>
> >***Point 3**: Additionally, beyond the overall evaluation results, I would like to see specific results across different domain and topics (as shown in Table 2). This could lead to new discoveries and spark more valuable discussions (involving different knowledge systems).*
>
> >**Response**: Thanks for the suggestion. Fortunately, we have all the performances. The model response as well as test accuracies are in the supplementary file. The reason why we do not include them in the paper is that diagrams in different domains vary quite differently, and the number of diagrams in each domain is a bit small. Thus, the test accuracy in each domain could be biased, and while presenting them, we might need to be more careful. Thus, we only report the overall accuracies to make sure the results are reliable. We will publish all the data and encourage future work for further exploration.
>
> ---
>
> >***Point 4**: The Chain of Thought approach mentioned by the authors is relatively simple, as it only involves adding prompts like “think Step-by-Step.” Are there any more effective ways to use Step-by-Step prompts for this kind of chart? I suggest that the authors explore this in greater depth.*
>
> >**Response**: Regarding the CoT setting, we follow the popular setting proposed by Wei et al., 2022 and Kojima et al., 2023. We agree that proposing a better CoT prompt is interesting but this is not our focus.
>
> ---
>
> >***Point 5**: However, for some simple relational charts, how can we be certain that these large models have not encountered similar charts during training? This point remains uncertain.*
>
> >**Response**: We also generate synthetic data on our own for our evaluations. Thus, we can reasonably assume that these diagrams are not seen by large models.
>
> ---
>
> >***Point 6**: By the way, in this work, beyond the evaluations conducted by the authors, I would like to see a model proposed by the authors specifically for understanding basic relational charts.*
>
> >**Response**: Our work mainly focuses on the evaluation and interpretation side. Besides, we focus on the general diagram instead of specific charts (see section 2.1 for the diagram definition). We agree that exploring the model improvement is interesting but this is not our focus. We would like to encourage future work to do that.
>
> **In summary, we appreciate the reviewer's effort in reviewing our work. But we would like to kindly ask the reviewer to re-evaluate our work based on our response.**

---

### Official Review · Reviewer_Y6Hj · 2024-11-02

**Soundness:** 4
**Presentation:** 4
**Contribution:** 3
**Rating:** 8
**Confidence:** 3

**Summary:**

The presented work examines to what degree large vision-language models (LVLM) understand diagrams. A test suite is developed that includes diagram-understanding questions for a synthetic and real-world dataset. The questions are distributed into four categories: Every question is either a recognition or reasoning question, and every question is either knowledge-required or knowledge-free. The questions are either applied to entities in the diagram or to relations between them. While LVLMs can recognize and reason about entities in synthetic diagrams, they show poor performance for entities on real-world data. Surprisingly, their performance improves on more complex, real-world diagrams. The authors provide evidence that this performance leap originates from semantic background knowledge that the LVLM bears. In a case study, the authors show that the LVLMs hallucinate some answers due to their semantic background knowledge.

**Strengths:**

- The paper is written well, and the figures help the reader understand the main ideas.
- The categorization of questions is intuitive and exemplified, and the test suite allows for further research in the direction of diagram understanding. An extensive appendix supports the main text by providing additional information about the curated test suite and the research methodology.
- The findings are novel and counter-intuitive. The semantic background leakage is well-induced and confirmed by extensive experiments. It highlights a surprising weakness of state-of-the-art LVLMs that should be considered when using them.
- The findings bear opportunities for future research. Semantic background leakage may also appear in other areas besides diagram understanding. Furthermore, it should be researched how LVLMs can be trained to be more robust and effective in diagram understanding.

**Weaknesses:**

The case study (Section 5.2) could be more extensive, and its design includes some shortcomings:
- The evaluation of the middle figure needs to be revised, as the correct answers are not included in the answer options. I propose to include an additional experiment, possibly in the appendix, with correct answer options. It would be highly interesting to observe if LVLMs even deviate from correct reasoning due to their inherent semantic background knowledge.
- Another important aspect that needs to be considered in the case study is the spatial positioning of the entities. To exclude the influence of spatial positionings on the observed hallucinations, I propose to rerun the case study with swapped spatial positions. Is the "fish" consistently predicted if swapped with the other entities?

**Questions:**

The main text does not state how the real-world diagram set was curated. Specifically, it would be interesting which filtering criteria were applied. Did you filter diagrams only by their domain? Did you apply a limit for the maximum number of nodes/relations/edge crossing?

---

> ### Author Response · Authors · 2024-11-13
> **Response to Review Y6Hj**
>
> We highly appreciate the reviewer for recognizing our test suite and admiring our findings for the community. Let us try to answer the proposed questions point by point below.
>
> >***Point 1**: The evaluation of the middle figure needs to be revised, as the correct answers are not included in the answer options…*
>
> >**Response**: We select the real example from the dataset as the case study. In fact, we also tried the case with the correct option. The answer is still the same. We may include these results in the main paper or Appendix.
>
> ---
>
> >***Point 2**: …To exclude the influence of spatial positionings on the observed hallucinations, I propose to rerun the case study with swapped spatial positions. Is the "fish" consistently predicted if swapped with the other entities?*
>
> >**Response**: Indeed, in Appendix E.2.1 (Figure 5), we explore if the position would affect the model’s performance and the answer is yes. We suppose that there is position bias in real diagrams as well. But for your information, we tried several times manually with entities shuffled and the answer is still ``fish’’.
>
> ---
>
> >***Question 1**: The main text does not state how the real-world diagram set was curated. Specifically, it would be interesting which filtering criteria were applied. Did you filter diagrams only by their domain? Did you apply a limit for the maximum number of nodes/relations/edge crossing?*
>
> >**Response**: We determine the domains first and manually select all diagrams in that domain for annotation. We filtered out some low-quality (e.g., blurred or over-simple) diagrams. During the question annotation, we also removed several diagrams that are extremely hard to annotate. We mention this in the revised version (see red lines 232-235).

---

### Official Review · Reviewer_o1o6 · 2024-11-03

**Soundness:** 3
**Presentation:** 3
**Contribution:** 3
**Rating:** 6
**Confidence:** 2

**Summary:**

This paper evaluates three LVLMs on diagram understanding. The authors curated a synthetic + real diagram dataset and investigated LLM performance on entity and relation recognition. Results suggest that LVLMs may not be truly understanding diagrams.

**Strengths:**

- The writing is very clear. As a reader, I feel that the hypotheses, experiment designs, and results are all clearly conveyed. I especially like how the authors layer experiments based on previous results to provide deeper and deeper insights.
- Experiments are overall well-designed. The tasks seem reasonable. The combination of synthetic and real diagrams is a strength that allows controllability and real-world validity. I like how the authors pay attention to details like varying the sizes and colors of arrows in the diagrams.

**Weaknesses:**

1. Making counterfactual variations of diagrams and asking LVLMs about them is certainly interesting, but I think it is not surprising that this should degrade LVLM performance. When strong prior is present and the diagram contradicts that, the LVLM could simply get confused. In such cases I think it is important to test if explicitly including instructions to ignore prior knowledge and solely answer using the information in the diagram improves the performance. Take Section 5.2 as an example. When no link is present (middle panel), asking "How many food chains are there?" sounds more like the goal is to test relevant ecology knowledge instead of diagram reading, and I think the LVLM's decision to hallucinate connections is actually warranted. In other words, this case in particular feels like an unfair trick question to me.
2. This paper tested 3 LVLMs. Perhaps testing a few more would be helpful, e.g., Claude 3.5 Sonnet.
3. Even though I appreciate the inclusion of a real-world dataset, it is exclusively focused on science. A broader scope would be better.

**Questions:**

- "Furthermore, we demonstrate that the models primarily rely on knowledge shortcuts when answering complex diagram reasoning questions." I am uneasy with the use of "primarily". The difference between KR and KF is at most ~15%. Saying "primarily" seems wrong to me. Saying knowledge is a shortcut LVLMs exploit seems reasonable, but could the authors justify why they said "primarily"?

---

> ### Author Response · Authors · 2024-11-13
> **Response to Review o1o6**
>
> Thanks for the positive comments! We are grateful that our experiment design is admired by the reviewer. It’s our great pleasure to provide useful and reliable insights for the community. Let us address the proposed concerns point by point below.
>
>
> >***Point 1**: …I think it is important to test if explicitly including instructions to ignore prior knowledge and solely answer using the information in the diagram improves the performance…*
>
> >**Response**: We appreciate this constructive and delicate suggestion! Indeed, it is intuitive that the model performance drops when there is a contradiction between input content and prior knowledge. But in all our experiments, we ask the model to focus on the provided diagrams (see lines 251-252 and red lines 517-518) to answer questions, which can alleviate the contradiction. We improve our presentation to make this clear in the revised version (line 518).
> >
> > Furthermore, we would like to kindly argue that in our scenario,  it is not reasonable to look at the diagram for the entities and give the answer based on prior knowledge (hypothesized model’s manner). Specifically, it’s worth noting that we design questions that can only be answered by relying on diagram images, where the only text input is the question (with answer options). Regarding the example in section 5.2, we ask "How many food chains are there?". If the model ignores the diagram, the answer can’t be given even with prior knowledge. If the model gives the answers based on the diagram, the answer is naturally from the diagram instead of the prior knowledge. The question is not about picking from two contradictory options. It is about answering questions by analyzing the input, i.e., the diagram. Thus, we regard the hypothesized model’s manner as unsuitable.
>
> ---
>
> >***Point 2**: This paper tested 3 LVLMs. Perhaps testing a few more would be helpful, e.g., Claude 3.5 Sonnet.*
>
> >**Response**: In our experiments, we find that all three models have similar performance manners. Thus, exploring additional models would cost much more money but probably provide similar results. Besides, when we started our experiments, Claude, as well as other LVLMs (e.g., Molmo, LLaMA3.2-Vision), were not available (in many countries). Therefore, we encourage future work on broader explorations of models.
>
> ---
>
> >***Point 3**: Even though I appreciate the inclusion of a real-world dataset, it is exclusively focused on science. A broader scope would be better.*
>
> >**Response**: Thanks for the suggestion. As mentioned by Reviewer 6WSg, we add discussions of more types of diagrams e.g., charts in related work (see red lines 912-920). Since we focus on real diagrams that contain rich background knowledge, limited by existing datasets (e.g., AI2D), we mainly focus on the scope of science. We would like to leave more comprehensive explorations on the diagram scope for future work.
>
> ---
>
> >***Question 1**: …I am uneasy with the use of "primarily". The difference between KR and KF is at most ~15%. Saying "primarily" seems wrong to me…*
>
> >**Response**: Thanks for the suggestion! We improve the word usage (see red lines 537-539) and will keep polishing the paper further.

---

> ### Comment · Reviewer_o1o6 · 2024-11-25
>
> Thank you for the response. It seems that there is a misunderstanding of my comments. I did not advocate for "look[ing] at the diagram for the entities and giv[ing] the answer based on prior knowledge". What I am saying is quite the contrary--that the model should be even more explicitly instructed to rely solely on the diagram to answer the question, without considering its prior knowledge. Without this explicit instruction, when the question resembles a trick question, it is imo not a fair assessment.
> The authors seem to have only made minor changes, so I will keep my score. That said, I still believe it is marginally *above* the acceptance threshold.

---

> > ### Author Response · Authors · 2024-11-25
> > **Thanks for the clarification**
> >
> > Thank you for the reply. We are sorry about the previous misunderstanding. Let us present the new results to you to see if they can address your main concern. We have also included these results in the **Appendix E.3.1** of the paper.
> >
> > ---
> >
> > Below, we provide some new results to show that **including instructions to ignore prior knowledge and solely answer using the information in the diagram provides the SAME or SLIGHTLY WORSE performance**.
> >
> > In our previous setting, our original system prompt was:
> >
> > > You are a visual assistant answering multiple choice questions about diagrams. Read the question, inspect the diagram, and answer with the correct choice in the following format: 'A) 0'.
> >
> > As requested by the reviewer, we design the prompt (to ignore the prior knowledge) as:
> >
> > > You are a visual assistant answering multiple choice questions about diagrams. Read the question, **only** inspect the diagram **but do not use your prior knowledge**, and answer with the correct choice in the following format: 'A) 0'.
> >
> > We select GPT-4o (under the CoT setting) that can provide fastest and best responses, and evaluate it on both real and synthetic diagrams (we skip entity-related questions on synthetic diagrams since the accuracies are already quite high, i.e., > 95%). Results are listed below (as well as in Appendix E.3.1).
> >
> > | Overall accuracy | Original prompt | New prompt (ignore the prior knowledge) |
> > | -------- | ------- | ------- |
> > | $Q_S(E\|\text{KF, NR})$  | 76.58 | 76.16 |
> > | $Q_S(E\|\text{KF, NC})$  | **70.15** | **62.74** |
> > | | | |
> > | $Q_R(V\|\text{KF, NR})$  | 93.60 | 95.30 |
> > | $Q_R(V\|\text{KR,NR})$  | 93.10 | 93.79 |
> > | $Q_R(V\|\text{KF, NC})$  | 79.05 | 79.24 |
> > | $Q_R(V\|\text{KR, NC})$ | **82.29** | **77.58** |
> > | $Q_R(E\|\text{KF, NR})$  | 73.90 | 75.41 |
> > | $Q_R(E\|\text{KR, NR})$  | 84.10 | 85.81 |
> > | $Q_R(E\|\text{KF, NC})$  | 65.00 | 66.50 |
> > | $Q_R(E\|\text{KR, NC})$  | 72.89 | 71.08 |
> >
> > ---
> >
> > **We would like to thank again for the reviewer’s reply and kindly ask if these new results address your main concern.**
> >
> > Additionally, **we are working on extending our experiments with Claude. But due to the rate limit we may not be able to update the results before the discussion period ends.**
> >
> > Best,
> >
> > The authors

---

> > > ### Author Response · Authors · 2024-12-02
> > > **Last day reminder**
> > >
> > > Dear Reviewer o1o6,
> > >
> > > Thank you once again for the time and effort you have dedicated to reviewing our paper! We are wondering if our last comment and latest updates addresses your questions. Thank you once again for your consideration!
> > >
> > > Best,
> > > The authors

---

> > > > ### Comment · Reviewer_o1o6 · 2024-12-03
> > > >
> > > > Thank you for the response. It seems that a slight change in the prompt following my suggestion results in better results in 6 settings and worse results in 4 settings. Therefore, I am still a bit uneasy over this particular conclusion. The updated system prompt now instructs the model to "only inspect the diagram but do not use your prior knowledge", which may have been too strong in advising the model not to rely on prior knowledge. One could imagine more tender ways of phrasing this, such as "when the diagram shows information contrary to your prior knowledge, defer to the information shown in the diagram". In any case, I think a more rigorous approach would be to carve out a small dev set on which to perform a bit of prompt engineering and before proceeding with the whole corpus. In light of this, I would like to keep my original "marginally above threshold" score.

---

### Official Review · Reviewer_6WSg · 2024-11-04

**Soundness:** 3
**Presentation:** 2
**Contribution:** 2
**Rating:** 5
**Confidence:** 4

**Summary:**

The paper demonstrates an in-depth evaluation if Vision Language models (VLM) can understand visual digram. The authors show the results both on their synthetically generated dataset as well as real visual diagrams curated from real datasets. They curate an extensive list of possible questions to evaluate VLM’s separately on each question category.

**Strengths:**

The template of questions used in the paper is quite extensive. The authors also carefully evaluated each template separately showing a holistic evaluation framework. Specifically, the key observation noticed under Q1 seems interesting to me. The ability of VLM to understand and reason well about entities while struggling with relationships shows that relationships are hard to decode in general. The performance gap between real and synthetic datasets is also interesting.

**Weaknesses:**

1. The paper write-up could perhaps be revisited. It is difficult to read Table 6 the first time — the relative improvement could perhaps be presented more intuitively.
2. The motivation behind using ‘knowledge as a shortcut' in sections 4 and 5 was not clearly stated. Was there a chance for choosing to construct a separate knowledge graph of textual content rather than labeling the visual entities in the original diagram? Providing the rationale behind this construction would be interesting. Since ‘knowledge’ is a quite generic word, it might be useful to define more precisely what kind of knowledge they are referring to early in the paper.

3. There are some relevant works from Chart, Graph, and 3D scene understanding that will be worth mentioning in the related works section:

        - ChartQA: https://arxiv.org/abs/2203.10244

        - Talk like a graph: Encoding graphs for large language models: https://research.google/blog/talk-like-a-graph-encoding-graphs-for-large-language-models/

        - CLEVR: A Diagnostic Dataset for Compositional Language and Elementary Visual Reasoning: https://openaccess.thecvf.com/content_cvpr_2017/html/Johnson_CLEVR_A_Diagnostic_CVPR_2017_paper.html

**Questions:**

1. Could it be possible to make the synthetic dataset public so that the reviewers have a better sense of its content?
2. The authors classified the diagram complexity based on the number of entities. Was there a reason they did not consider the number of relationship to measure complexity?

---

> ### Author Response · Authors · 2024-11-13
> **Response to Review 6WSg**
>
> We thank the reviewer for recognizing our extensive and holistic experiment design. We explore whether LVLMs understand visual language by answering two questions in the Introduction with a series of experiments. We include carefully designed synthetic diagrams and comprehensive real diagrams from multiple domains to provide reliable insights. We try to address the proposed concerns point by point below.
>
> >***Point 1**: The paper write-up could perhaps be revisited. It is difficult to read Table 6 the first time — the relative improvement could perhaps be presented more intuitively.*
>
> >**Response**: We improved the presentation of all tables in our paper by adding color to differentiate the zero-shot performance and CoT performance. Besides, we changed the table headers and descriptions of Tables 6, 7, and 8 to make them easier to follow.
>
> ---
>
> >***Point 2**: The motivation behind using ‘knowledge as a shortcut' in sections 4 and 5 was not clearly stated…*
>
> >**Response**: Our paper is organized by answering the two questions in the Introduction. Specifically, we use one intuition, five observations, and one finding to connect all experiments. The motivation behind knowledge shortcuts is stated after observation 3 as well as at the beginning of section 4. We emphasize these transition contents to make them easier to follow (red lines 362-363). The motivation is also pasted below for your information.
> >
> >> We substantiate that LVLMs cannot understand relations in synthetic diagrams, yet this finding reveals a contradiction. While the models do not inherently possess the ability to recognize relations, they are able to do so in real diagrams. This outcome contradicts our initial intuition (Intuition 1). Therefore, we further investigate these counterintuitive findings by examining the key difference between synthetic and real diagrams: the role of background knowledge.
>
> ---
>
> >***Point 3**: Since ‘knowledge’ is a quite generic word, it might be useful to define more precisely what kind of knowledge they are referring to early in the paper.*
>
> >**Response**: We will further polish our word usage. “knowledge’’ here refers to the background knowledge (e.g., commonsense knowledge) that can be used to understand the diagram.
>
> ---
>
> >***Point 4**: There are some relevant works from Chart, Graph, and 3D scene understanding that will be worth mentioning in the related works section...*
>
> >**Response**: Thanks for the references. We add the discussion of these works in our related work (see red lines 910-926).
>
>
> ---
>
> >***Question 1**: Could it be possible to make the synthetic dataset public so that the reviewers have a better sense of its content?*
>
> >**Response**: We attached all materials in the supplementary, including the code, data, and model responses. The synthetic dataset used in our experiment is also contained. Besides, we provide examples (including diagrams and responses) in the Appendix (Figure 10-33) for each experiment.
>
> ---
>
> >***Question 2**: The authors classified the diagram complexity based on the number of entities. Was there a reason they did not consider the number of relationship to measure complexity?*
>
> >**Response**: Thanks for this good question! For real diagrams, the number of entities often increases with the number of relations. Thus, for simplicity, we use the entity number to roughly measure the diagram complexity. We could also report the results by the number of relations if the reviewer prefers.

---

> > ### Author Response · Authors · 2024-12-02
> > **Last day reminder**
> >
> > Dear Reviewer 6WSg,
> >
> > Thank you once again for the time and effort you have dedicated to reviewing our paper! We have carefully revised the manuscript in response to the feedback and included a response addressing all the raised points.
> >
> > Consider that you indicate **strong confidence** about the review evaluation. Please let us know if your concerns and questions have been addressed. If so, we are grateful **if you are willing to re-evaluate our work and adjust the score accordingly**.
> >
> > We are more than happy to address any additional questions or feedback you may have today. Thank you once again for your consideration!
> >
> > Best,
> >
> > The authors

---

> ### Comment · Reviewer_6WSg · 2024-12-02
> **Response to Authors**
>
> Thanks to the authors for their response. I appreciate the time they invested in it. I want to stay with my original rating since I still feel it may not be an adequate contribution to be considered as a full paper.

---

### Meta-Review · Area_Chair_RDMk · 2024-12-20

**Metareview:**

The paper presents an evaluation of whether VLMs can understand visual diagrams. While VLMs recognize and reason about entities, they struggle with reasoning about relations in synthetic diagrams. Surprisingly, their performance on understanding relations improves on more complex, real-world diagrams. The authors provide evidence that this performance leap might not be due to visual understanding, but due to pre-trained knowledge baked into these language models

Most Reviewers found the paper to be clearly written, experiments well-designed with useful insights. However, some concerns were raised about sensitivity to prompts. One of the suggestions was to conduct a more thorough investigation of prompt-engineering approaches on a dev-set, before proceeding to the whole corpus. During the reviewer discussion period, reviewers felt that the paper can benefit from advanced prompting techniques, or fine-tuning open-weights model to be considered a full paper. While proposing a new model maybe out of scope, additional experiments with open-weights model as well as fine-tuning on synthetic dataset would have significantly increased the contributions of the paper. It would help answer questions about how much data is needed to improve diagram understanding, and what kind of data is required. Such insights will be useful to the broader LLM/ VLM community.

**Additional Comments On Reviewer Discussion:**

1. Citing relevant works -- Reviewers pointed out some relevant works that were added to the paper.

2. Analysing prompt template design -- During rebuttal, the authors conducted experiments with an additional prompt template and found that the model is sensitive to these choices. Reviewer o1o6 rightly pointed out that further analysis is needed on a dev-set before running experiments on the whole corpus.

3. Reviewer TikE mentioned that beyond the evaluations, they would like to see a model proposed to improve diagram understanding. The authors felt that this is not the focus of the proposed work, and is future work. Like mentioned above, while proposing a new model maybe out of scope, additional experiments with open-weights model, and potentially fine-tuned versions of those models will certainly strengthen the contributions of the paper.

---

### Decision · Program_Chairs · 2025-01-22

Reject